# Endosomal chemokine receptor signalosomes regulate central mechanisms underlying cell migration

**Hyunggu Hahn[1,2†‡], Carole Daly[3†‡], John Little IV[1,4†‡], Nicole A Perry-Hauser[4], Emmanuel Flores-Espinoza[1,2], Asuka Inoue[5,6], Bianca Plouffe[3*], Alex RB Thomsen[1,2,4*]**

[1]Department of Molecular Pathobiology, New York University College of Dentistry, New York, United States; [2]NYU Pain Research Center, New York University College of Dentistry, New York, United States; [3]Wellcome-Wolfson Institute for Experimental Medicine, Queen's University Belfast, Belfast, United Kingdom; [4]Department of Surgery, Columbia University Columbia University Vagelos College of Physicians and Surgeons, New York, United States; [5]Graduate School of Pharmaceutical Science, Tohoku University, Sendai, Japan; [6]Graduate School of Pharmaceutical Science, Kyoto University, Kyoto, Japan

*For correspondence:
b.plouffe@qub.ac.uk (BP);
art8@nyu.edu (ARBT)

†These authors contributed equally to this work

‡Co-first authors

## eLife Assessment

This is an **important** study that provides CCR7-APEX2 proximity labelling mass spectrometry data that is expected to provide new insights into CCR7 signaling partners and pathways. The study is technically easy to follow and the data is **convincing**. It will be interesting in the future to have complementary studies in lymphocytes/dendritic cells that endogenously express CCR7. This is of value to the community, and there are likely multiple opportunities to use the APEX2 data set to extend these findings, strengthen some claims, and even explore a new pathway identified in the APEX2 data set.

**Abstract** Chemokine receptors are GPCRs that regulate the chemotactic migration of a wide variety of cells including immune and cancer cells. Most chemokine receptors contain features associated with the ability to stimulate G protein signaling during β-arrestin-mediated receptor internalization into endosomes. As endosomal signaling of certain non-GPCR receptors plays a major role in cell migration, we chose to investigate the potential role of endosomal chemokine receptor signaling on mechanisms governing this function. Applying a combination of pharmacological and cell biological approaches, we demonstrate that the model chemokine receptor CCR7 recruits G protein and β-arrestin simultaneously upon chemokine stimulation, which enables internalized receptors to activate G protein from endosomes. Furthermore, spatiotemporal-resolved APEX2 proteome profiling shows that endosomal CCR7 uniquely enriches specific Rho GTPase regulators as compared to plasma membrane CCR7, which is directly associated with enhanced activity of the Rho GTPase Rac1 and chemotaxis of immune T cells. As Rac1 drives the formation of membrane protrusions during chemotaxis, our findings suggest an important integrated function of endosomal chemokine receptor signaling in cell migration.

## Introduction

Chemokines are small proteins secreted from locations of physiological damage such as infection and inflammation. From these locations a gradient of chemokines is generated that attracts immune cells by activating their cell surface chemokine receptors, triggering cell migration and differentiation to combat the pathophysiological state. The chemokine system is also taken advantage of by certain cancer cells including melanoma, glioblastoma, prostate, gastric, pancreatic, esophageal, ovarian, lung, colorectal, and breast cancer (*Sarvaiya et al., 2013*). In malignant cells, upregulation of chemokine receptors allows them to invade and metastasize to locations distinct from their origin and from where specific chemokines are secreted (*Sarvaiya et al., 2013*).

There exist 23 chemokine receptors that belong to the highly druggable superfamily of G protein-coupled receptors (GPCRs). Canonically, GPCRs activate heterotrimeric G proteins (Gαβγ) at the cell surface, causing downstream signaling throughout the cell (*Gilman, 1987*). This initial phase of G protein signaling is terminated via a specialized desensitization mechanism that includes phosphorylation of receptors by GPCR kinases and subsequent recruitment of β-arrestins (βarrs) to the phosphorylated receptor (*Moore et al., 2007*). βarrs engage receptors at an overlapping transmembrane core region to where G proteins bind, and thus βarr recruitment to the receptor sterically blocks further G protein activation (*Lohse et al., 1990*; *Lee et al., 2020*). In addition, βarrs promote the internalization of GPCRs into endosomes, thereby removing them from the source of the activating ligand (*Goodman et al., 1996*).

In more recent times work has shown that some GPCRs continue to activate G proteins following βarr-mediated internalization into endosomes (*Jensen et al., 2017*; *Thomsen et al., 2018*; *Thomsen et al., 2016*; *Irannejad et al., 2013*; *Calebiro et al., 2009*; *Wehbi et al., 2013*; *Ferrandon et al., 2009*; *Mullershausen et al., 2009*; *Flores-Espinoza and Thomsen, 2024*). As G protein and βarr association to GPCRs have historically been considered mutually exclusive events, endosomal G protein signaling is difficult to reconcile within the general understanding of GPCR signaling. However, our recent work showed that certain GPCRs containing serine/threonine phosphorylation site clusters in their carboxy-terminal tail bind to βarrs in a specific conformation termed 'tail conformation' (*Cahill et al., 2017*; *Kumari et al., 2016*; *Kumari et al., 2017*). In this conformation, βarr only interacts with the receptor carboxy-terminal tail thereby permitting the receptor transmembrane core to bind to G proteins simultaneously to form a GPCR–G protein–βarr complex or 'megaplex' (*Thomsen et al., 2016*; *Cahill et al., 2017*; *Nguyen et al., 2019*). The formation of these megaplexes highlights how certain receptors that form strong associations with βarrs can continue to stimulate G protein signaling over prolonged periods of time after they have been internalized into endosomes by βarrs. Despite having established a mechanistic understanding underlying endosomal G protein signaling by internalized GPCRs, knowledge about its functional role remains minimal.

Most chemokine receptors couple to similar G protein subtypes (*Hughes and Nibbs, 2018*) and contain serine/threonine clusters in their carboxy-terminal tail (*Table 1*), which raises the possibility that the formation of megaplexes and endosomal G protein signaling are general mechanisms applied by these receptors to influence physiological responses such as cell migration. Interestingly, endosomal signaling by other non-GPCR receptor systems such as receptor tyrosine kinases have a profound role in cell migration and control transport of various signaling complexes and other proteins between the cell surface and endosomal compartments during this process (*Palamidessi et al., 2008*; *Schiefermeier et al., 2011*). Therefore, our work here tests the hypothesis that chemokine receptors can stimulate G proteins from endosomes to regulate common signaling networks or 'signalosomes' that direct cell migration.

## Results

### CCR7-induced G protein signaling is unaffected by βarr recruitment and βarr-mediated receptor endocytosis

To study the potential role of endosomal chemokine receptor signaling we used the model receptor CCR7, which is expressed by several subsets of immune cells and regulates their homing to the lymph nodes (*Förster et al., 2008*). This receptor is an ideal model receptor to study endosomal G protein signaling as it is known to be active at endomembranes (*Laufer et al., 2019*) and is stimulated by two endogenous chemokines CCL19 and CCL21 equally effective at 100 nM (*Kohout et al., 2004*).

**Table 1.** Schematic overview of the C-terminal tail region of common chemokine receptors.
Potential serine/threonine phosphorylation sites are colored red, and serine/threonine clusters are underlined in bold.

| Chemokine receptor | Chemokines | C-terminal tail |
|---|---|---|
| CCR1 | CCL3, CCL4, CCL5, CCL7, CCL8, CCL13, CCL14, CCL16, CCL23 | RVAVHLVKWLPFLSVDRLERVSSTSPSTGEHELSAGF |
| CCR2A/B | CCL2, CCL7, CCL8, CCL11, CCL13, CCL16, CCL24, CCL26 | A:CRIAPLQKPVCGGPGVRPGKNVKVTTQGLLDGRGKGKSIGRAPEASLQDKEGA B:KQCPVFYRETVDGVTSTNTPSTGEQEVSAGL |
| CCR3 | CCL2, CCL5, CCL7, CCL8, CCL11, CCL13, CCL15, CCL24, CCL26, CCL28, CXCL9, CXCL10, CXCL11 | HLLMHLGRYIPFLPSEKLERTSSVSPSTAEPELSIVF |
| CCR4 | CCL17, CCL22 | TCRGLFVLCQYCGLLQIYSADTPSSSYTQSTMDHDLHDAL |
| CCR5 | CCL2, CCL3, CCL4, CCL5, CCL7, CCL8, CCL11, CCL13, CCL14, CCL16 | KHIAKRFCKCCSIFQQEAPERASSVYTRSTGEQEISVGL |
| CCR6 | CCL20, β-defensin 4 A | CVRRKYKSSGFSCAGRYSENISRQTSETADNDNASSFTM |
| CCR7 | CCL19, CCL21 | DLGCLSQEQLRQWSSCRHIRRSSMSVEAETTTTFSP |
| CCR8 | CCL1, CCL8 | SCSQIFNYLGRQMPRESCEKSSSCQQHSSRSSSVDYIL |
| CCR9 | CCL25 | QWVSFTRREGSLKLSSMLLETTSGALSL |
| CCR10 | CCL27, CCL28 | GGSCPSGPQPRRGCPRRPRLSSCSAPTETHSLSWDN |
| CXCR1 | CXCL6, CXCL8 | GLVSKEFLARHRVTSYTSSSVNVSSNL |
| CXCR2 | CXCL1, CXCL2, CXCL3, CXCL5, CXCL6, CXCL7, CXCL8 | GLISKDSLPKDSRPSFVGSSSGHTSTTL |
| CXCR3 | CCL5, CCL7, CCL11, CCL13, CCL19, CCL20, CXCL9, CXCL10, CXCL11, CXCL12α | GCPNQRGLQRQPSSSRRDSSWSETSEASYSGL |
| CXCR4 | CXCL12, CXCL12α, CXCL12β, CXCL12γ, CXCL12δ, CXCL12ε, CXCL12φ | SVSRGSSLKILSKGKRGGHSSVSTESESSSFHSS |
| CXCR5 | CXCL13 | KLGCTGPASLCQLFPSWRRSSLSESENATSLTTF |
| CXCR6 | CXCL16 | CLPYLGVSHQWKSSEDNSKTFSASHNVEATSMFQL |
| XCR1 | ? | FWFCRLQAPSPASIPHSPGAFAYEGASFY |
| CX3CR1 | CX3CL1 | CLAVLCGRSVHVDFSSSESQRSRHGSVLSSNFTYHTSDGDALLLL |

Compared with the full agonist CCL19 that robustly stimulates $G_{i/o}$ signaling, receptor phosphorylation, βarr recruitment, and CCR7 internalization, the agonist CCL21 promotes full $G_{i/o}$ protein activation but only partial receptor phosphorylation, βarr recruitment, and internalization of CCR7 (*Kohout et al., 2004*; *Zidar et al., 2009*). Using the enhanced bystander bioluminescence resonance energy transfer (EbBRET)-based biosensors RlucII-βarr1/2 and cell surface-anchored rGFP-CAAX to monitor βarr1/2 recruitment to plasma membrane upon chemokine-stimulation in CCR7-expressing HEK293 cells (*Figure 1A*; *Namkung et al., 2016*), we confirmed that activating CCR7 by CCL19 leads to robust βarr1/2-recruitment, whereas CCL21 only promotes partial βarr1/2-recruitment responses (*Figure 1B–C*). Co-expressing RlucII-βarr1/2 and the endosomally-located rGFP-Rab5 in HEK293-CCR7 cells (*Figure 1A*; *Namkung et al., 2016*), CCL19 stimulation leads to similar strong βarr1/2 trafficking to endosomes whereas CCL21 stimulation of CCR7 only causes a minor translocation of βarr1/2 to the endosomes (*Figure 1B–C*). No enhanced EbBRET between RlucII-βarr1/2 and rGFP-CAAX/Rab5 was detected upon chemokine stimulation in HEK293 cells that do not express CCR7 (*Figure 1—figure supplement 1A–B*). These results suggest that CCR7 recruits and is internalized by βarrs robustly when stimulated with CCL19, but only to a modest degree by CCL21. Similar results were found when using β-galactosidase enzyme complementation-based DiscoverX assay to monitor direct βarr2 recruitment to CCR7 (*Figure 1—figure supplement 1C–D*).

Next, to investigate how CCL19/CCL21-stimulated βarrs recruitment affects CCR7 internalization in real-time, we applied a split NanoLuc (or NanoBiT) approach in which the small-BiT (SmBiT) is fused the carboxy-terminal of CCR7 and the large-BiT (LgBiT) is fused to the cell surface anchor CAAX. In resting state, the CCR7-SmBiT and LgBiT-CAAX are in subcellular proximity at the plasma membrane and form functional nanoluciferase enzymes, which catalyzes the conversion of coelenterazine h

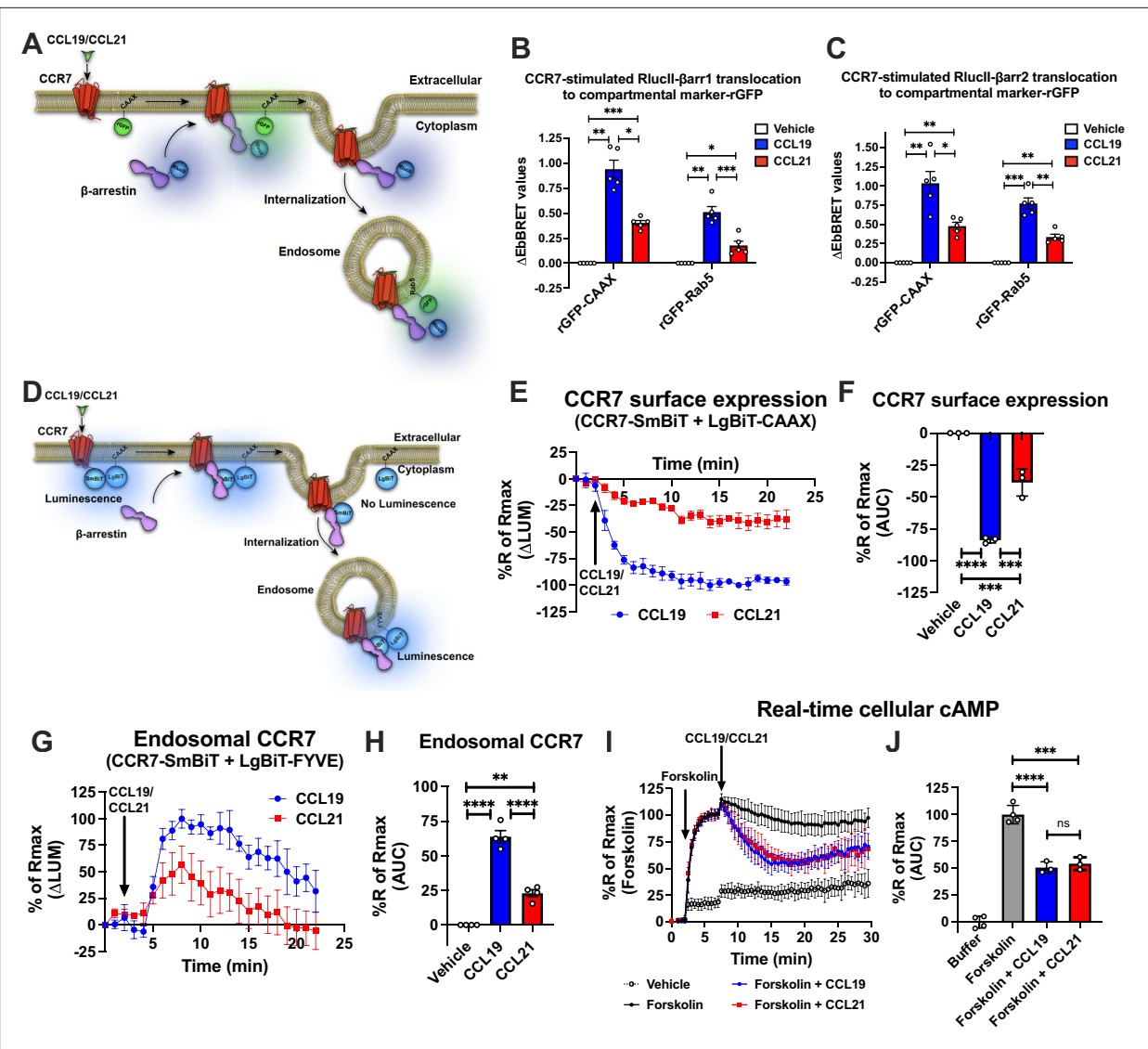

**Figure 1.** CCR7-stimulated β-arrestin (βarr) recruitment, receptor internalization, and G$_{i/o}$ signaling. (**A**) Schematic representation of the experimental design used to monitor enhanced bystander bioluminescence resonance energy transfer (EbBRET) between RlucII-βarr1/2 and rGFP-CAAX or RlucII-βarr1/2 and rGFP-Rab5 upon chemokine stimulation of CCR7. (**B–C**) EbBRET signal between (**B**) RlucII-βarr1 or (**C**) RlucII-βarr2 recruitment to plasma membrane-anchored rGFP-CAAX or early endosome-anchored rGFP-Rab5 in response to 100 nM CCL19, 100 nM CCL21, or vehicle control stimulation. Data represent the mean ± SE of N=5 experiments. (**D**) Schematic representation of the experimental design used to monitor CCR7 internalization by detecting loss of luminescence generated by CCR7-SmBiT and LgBiT-CAAX or endosomal CCR7 translocation by measuring gain of luminescence by CCR7-SmBiT and LgBiT-FYVE. (**E**) Change in luminescence signal generated between CCR7-SmBiT and LgBiT-CAAX in response to 100 nM CCL19 or 100 nM CCL21. The response to chemokine stimulation was normalized to vehicle control. (**F**) Area under the curve (AUC) was used to calculate the total internalization response for each chemokine ligand. Data represent the mean ± SE of N=3 experiments. (**G**) Change in luminescence signal generated between CCR7-SmBiT and LgBiT-FYVE in response to 100 nM CCL19 and 100 nM CCL21. The response to chemokine stimulation was normalized to vehicle control. (**H**) Area under the curve (AUC) was used to calculate the total internalization response for each chemokine ligand. Data represent the mean ± SE of N=4 experiments. (**I**) HEK293-CCR7 cells transiently expressing the real-time cAMP sensor CAMYEL were challenged with 10 µM forskolin (or vehicle buffer) to increase cAMP production. 5 min later, the cells were stimulated with 100 nM CCL19, 100 nM CCL21, or vehicle buffer, and inhibition of cAMP production was followed as an indirect measurement of G$_{i/o}$ activation. (**J**) AUC was used to calculate the total cAMP for each chemokine ligand. Data represent the mean ± SE of N=3–4 experiments. (F, H, and J) One-way ANOVA or (**B and C**) two-way ANOVA with (**B, C, F, H**) Tukey's or (**J**) Sidak's multiple comparison post hoc tests were performed to determine statistical differences between the distinct conditions (*p<0.05; **p<0.01; ***p<0.001; ****p<0.0001).

The online version of this article includes the following figure supplement(s) for figure 1:

**Figure supplement 1.** Negative control experiments in HEK293 cells and verification of CCL19/CCL21-induced βarr2 recruitment to CCR7 using the DiscoverX assay.

resulting in the emission of a bright luminescence signal (*Figure 1D*). Upon activation and internalization of CCR7, the CCR7-SmBiT is removed from the cell surface-located LgBiT-CAAX which results in a decrease in the luminescence signal (*Figure 1D*). Strong and immediate reduction in this signal was observed when the cells were stimulated with CCL19 (*Figure 1E–F*). In contrast, CCL21-stimulated CCR7 only led to a modest decrease in signal suggesting that CCL19-stimulation leads to robust CCR7 internalization whereas most CCR7 stays at the plasma membrane upon CCL21 activation (*Figure 1E–F*). In addition to this approach, we also measured the chemokine-stimulated trafficking of CCR7-SmBiT to the early endosomal FYVE-LgBiT marker (*Figure 1D*). In this case, CCL19 stimulation led to a robust increase in luminescence whereas CCL21 activation only led to a partial increase in the signal (*Figure 1G–H*). Interestingly, the kinetic patterns of the signals indicate that the trafficking of CCR7 to endosomes was slower and more transient as compared to its removal from the cell surface, which could be a result of further trafficking of the receptor to late or recycling endosomes as well as the trans-Golgi network (TGN; *Figure 1G–H*). Moreover, CCL19/CCL21-stimulation did not promote internalization or trafficking to endosomes of the vasopressin type 2 receptor ($V_2R$)-SmBiT construct validating that these chemokines act specifically via the CCR7-SmBiT system (*Figure 1—figure supplement 1E–F*).

Finally, to investigate how βarrs recruitment and receptor internalization affect G protein signaling in real-time, we applied the cAMP sensor CAMYEL (*Jiang et al., 2007*) to monitor $G_{i/o}$ activity in HEK293-CCR7 cells. In contrast to βarrs recruitment and receptor internalization, CCL19 and CCL21 challenge inhibited forskolin-induced cAMP production to a similar extent in HEK293-CCR7 cells (*Figure 1I–J*), but had no effect on the cAMP response in mock-transfected HEK293 cells (*Figure 1—figure supplement 1G–H*). This supports previous observations, which suggest that both CCL19 and CCL21 act as full agonists in stimulating $G_{i/o}$ signaling (*Kohout et al., 2004*; *Zidar et al., 2009*). Interestingly, this $G_{i/o}$ signaling was sustained throughout the experiment and not acutely desensitized by βarrs recruitment and/or receptor internalization as other receptor systems (*Figure 1I–J*; *Violin et al., 2008*).

## CCR7 engages with βarr1 and $G_{i/o}$ protein simultaneously upon chemokine stimulation

Surprisingly, neither the degree of βarr recruitment nor receptor internalization appears to affect CCR7-mediated G protein signaling as expected for most GPCRs. As we recently found a correlation between the presence of phosphorylation site clusters in GPCRs and their ability to form GPCR–G protein–βarr megaplexes (*Thomsen et al., 2016*), here we hypothesized that CCL19-CCR7 stimulates $G_{i/o}$ while being internalized into endosomes by βarrs (*Figure 2A*). As numerous $G_{i/o}$-coupled GPCRs recently were demonstrated to co-couple with G protein and βarrs, this hypothesis seems plausible (*Smith et al., 2021*). In contrast, since the CCL21 challenge only leads to minor CCR7 internalization, most of the CCL21-CCR7-mediated $G_{i/o}$ activation takes place at the plasma membrane (*Figure 2A*).

Simultaneous GPCR association with G protein and βarr is facilitated by βarrs complexing exclusively through the phosphorylated receptor carboxy-terminal tail. We have previously demonstrated that other GPCRs with carboxy-terminal phosphorylation site clusters bind to βarrs in this tail conformation, and thus, it seemed likely that CCR7 could form them as well. To test this, we used a βarr1 mutant where the region of βarr1 that interacts with GPCR transmembrane cores, the fingerloop (FL), is deleted (βarr1-ΔFL). We showed previously using negative stain electron microscopy and single-particle averaging that βarr1-ΔFL almost entirely forms tail conformation complexes with GPCRs (*Cahill et al., 2017*). Using the EbBRET pair RlucII-βarr1-ΔFL and rGFP-CAAX (plasma membrane-anchored rGFP) in CCR7-expressing HEK293 cells, we probed the degree of βarr1-ΔFL recruitment to CCR7 in response to CCL19, CCL21, or vehicle. In addition, we tested how well βarr1-ΔFL internalizes with CCR7 into endosomes using the EbBRET pair RlucII-βarr1-ΔFL and rGFP-Rab5 (early endosome-anchored rGFP). Interestingly, both CCL19 and CCL21 stimulation led to the recruitment of βarr1-ΔFL to CCR7 at the membrane although CCL19 promoted this recruitment to a significantly higher degree than CCL21 (*Figure 2B*). Furthermore, CCL19 and CCL21 stimulation promotes βarr1-ΔFL translocation to endosomes (*Figure 2B*). No response to chemokine stimulation was observed in mock-transfected HEK293 cells that do not express CCR7 (*Figure 2—figure supplement 1A*). These results indicate that CCR7 can form tail conformation complexes with βarr1 in cells and that the stability of this complex is sufficiently strong for βarr1 to internalize the receptor into endosomes.

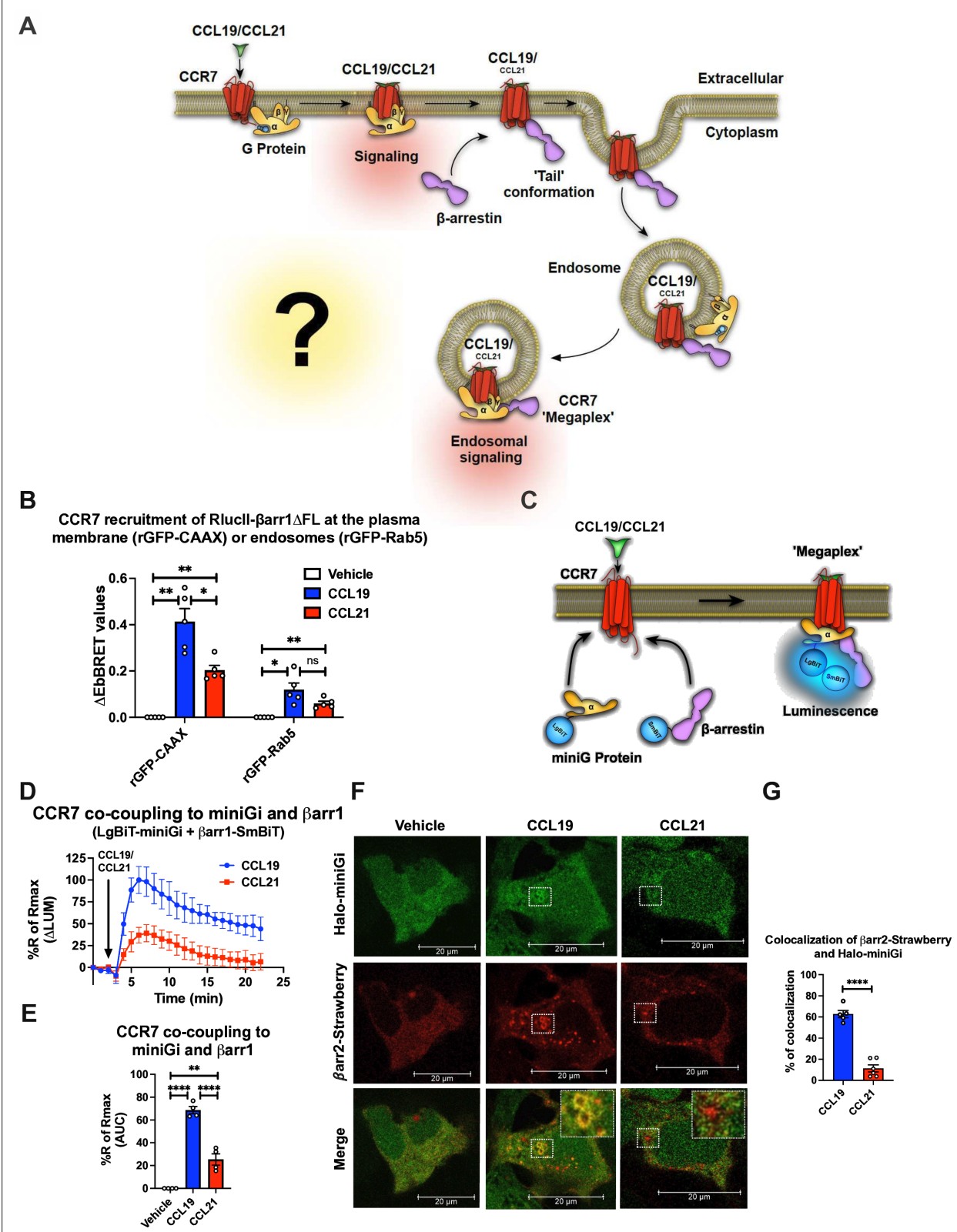

**Figure 2.** Chemokine-induced CCR7–$G_{i/o}$–β-arrestin (βarr) complex formation. (**A**) Schematic illustration of the working hypothesis. CCR7-mediated G protein signaling does not appear to be affected by βarr recruitment or receptor internalization. Therefore, we hypothesized that CCR7 associates and internalizes with βarr in the 'tail' conformation where βarr does not block the G protein-binding site within CCR7. As this site is available, CCR7 can interact simultaneously with G protein and βarr to form a CCR7–$G_{i/o}$–βarr megaplex, which enables the receptor to stimulate G proteins while being

*Figure 2 continued*

internalized into endosomes. (**B**) Enhanced bystander bioluminescence resonance energy transfer (EbBRET) signal between RlucII-βarr1ΔFL recruitment to plasma membrane-anchored rGFP-CAAX or endosomally-anchored rGFP-Rab5 in response to 100 nM CCL19, 100 nM CCL21, or vehicle control stimulation. Data represent the mean ± SE of N=5 experiments. (**C**) Schematic representation of the experimental design used to monitor luminescence upon proximity between SmBiT-βarr1 and LgBiT-miniG protein in response to CCR7 activation. (**D**) Change luminescence measured upon stimulation of HEK293-CCR7 cells expressing SmBiT-βarr1 and LgBiT-miniGi in response to 100 nM CCL19 or 100 nM CCL21 stimulation. The response to chemokine stimulation was normalized to vehicle control. (**E**) Area under the curve (AUC) was used to calculate the total response for each chemokine ligand. Data represent the mean ± SE of N=4 experiments. (**F**) Confocal microscopy imaging displaying HEK293 cells co-expressing, CCR7, βarr2-Strawberry and Halo-miniGi protein. In the experiment the cells were treated with either 100 nM CCL19, 100 nM CCL21, or vehicle control for 30 min. The images are representative from N=3 experiments. (**G**) Co-localization quantification analysis of βarr2-Strawberry and Halo-miniGi from six representative confocal microscopy images per condition. (**G**) Student's *t*-test, (**E**) one-way ANOVA, or (**B**) two-way ANOVA with Tukey's multiple comparison post hoc tests were performed to determine statistical differences between the distinct conditions (*$p<0.05$; **$p<0.01$; ***$p<0.001$; ****$p<0.0001$).

The online version of this article includes the following figure supplement(s) for figure 2:

**Figure supplement 1.** Control experiments to confirm that transducers are not recruited in HEK293 cells.

Since CCR7 forms tail conformation complexes with βarr1 upon agonist stimulation where the G protein binding site is available, we next sought to interrogate whether the receptor binds to βarrs and $G_{i/o}$ simultaneously (*Figure 2C*). To do this, we used a NanoBiT approach where SmBiT is fused to βarr1 and LgBiT is fused to the Gα subunit. If CCR7 recruits G protein and βarr1 simultaneously, the SmBiT-βarr1 and LgBiT-G will come into close proximity resulting in a luminescence signal. Instead of using the full-length Gα subunit for this approach as done previously (*Thomsen et al., 2016*), we used a truncated surrogate version commonly referred to as miniGi. MiniG proteins are engineered Gα subunits that have been developed for the major Gα subunit families ($G_s$, $G_{i/o}$, $G_{q/11}$, and $G_{12/13}$). Key modifications of these Gα subunits include: (1) a truncated N-terminal that deletes membrane anchors and Gβγ-binding surface; (2) deletion of the α-helical domain; (3) mutations that improve protein stability in vitro; and (4) a mutation in the C-terminal α5 helix that uncouples GPCR binding from nucleotide release, therefore, stabilizing receptor–miniG protein complexes in the presence of guanine nucleotides (*Wan et al., 2018*; *Nehmé et al., 2017*). These alterations also enable miniG proteins to report active receptor conformations in living cells by measuring their recruitment from the cytosol to GPCRs at different membrane compartments (*Wan et al., 2018*). As most membranes involved in GPCR trafficking contain endogenous G proteins (*Jang et al., 2024*), the presence of active miniG-coupling receptors at these subcellular compartments suggests that G proteins are stimulated from these sites.

The properties of miniG proteins are also beneficial when monitoring the simultaneous recruitment of SmBiT-βarr1 and LgBiT-miniG to GPCRs. Since both probes are expressed in the cytosol where they randomly collide, simultaneous translocation to the receptor in cell membranes will lead to a very specific increase in luminescence (*Figure 2C*). In contrast, using a wild-type full-length Gα subunit and βarr NanoBiT probes, which are located at membranes and cytosol, respectively, recruitment of βarrs to the receptor could lead to potential bystander effects as a result of βarrs-translocation from the cytosol to the plasma membrane.

In HEK293-CCR7 cells expressing SmBiT-βarr1 and LgBiT-miniGi, CCL19-stimulation leads to a robust increase in luminescence (*Figure 2—figure supplement 1B–C*). This signal was specific for miniGi as stimulation of CCR7 in cells expressing SmBiT-βarr1 and LgBiT-miniGs led to a small decrease in luminescence (*Figure 2—figure supplement 1B–C*). This reduction in basal luminescence might be caused by a reduction of random collision between SmBiT-βarr1 and LgBiT-miniGs upon CCR7 stimulation where SmBiT-βarr1, but not LgBiT-miniGs, translocates to the plasma membrane. These results indicate that CCL19-stimulated CCR7 recruits $G_{i/o}$ and βarrs simultaneously. Next, we compared the ability of CCL19 and CCL21 to provoke co-coupling of $G_{i/o}$ and βarrs to CCR7. Interestingly, both CCL19 and CCL21 stimulated co-coupling of $G_{i/o}$ and βarrs to CCR7, although CCL19 promoted the formation of these complexes to a significantly larger degree than CCL21 (*Figure 2D–E*). The response to chemokine stimulation was not observed in mock-transfected HEK293 cells (*Figure 2—figure supplement 1D*).

GPCR–G protein–βarr megaplexes have been described as a mechanism by which GPCRs that bind βarrs exceptionally well via the phosphorylated C-terminal tail can promote endosomal G protein signaling (*Thomsen et al., 2016*). Therefore, we investigated whether $G_{i/o}$ and βarrs are recruited

simultaneously to internalized CCR7 in endosomes using confocal microscopy. For this purpose, CCR7-expressing HEK293 cells co-expressing βarr2-Strawberry and Halo-miniGi were stimulated with either CCL19 or CCL21 for 30 min whereafter the subcellular location of these two probes was assessed by confocal microscopy. In this setup, CCL19-CCR7 recruited βarr2-Strawberry and Halo-miniGi to endosomal-shaped intracellular locations whereas this co-localization was minor for CCL21-CCR7 (*Figure 2F–G*). Together, these results indicate that CCL19 stimulation leads to robust endosomal and CCR7-mediated recruitment of $G_{i/o}$ and βarr1.

## Stimulation of CCR7 by CCL19 leads to robust activation of $G_{i/o}$ signaling from endosomes

To test whether the internalized CCR7 stimulates $G_{i/o}$ from endosomal compartments, we used the NanoBiT biosensor pair Rap1GAP-SmBiT and LgBiT-FYVE (*Figure 3A*). Rap1GAP is an effector protein that specifically binds to active $G_{i/o}$, but not to inactive $G_{i/o}$ (*Avet et al., 2022*), and thus, endosomal $G_{i/o}$ activation can be detected by measuring the recruitment of Rap1GAP-SmBiT to the endosomal marker LgBiT-FYVE (*Latorre et al., 2022*). Using this approach, we observed that both CCL19 and CCL21 stimulate the recruitment of Rap1GAP-SmBiT to LgBiT-FYVE in HEK293-CCR7 cells (*Figure 3B–C*). Similar to CCR7 internalization, CCL19 facilitated this response to a significantly higher degree as compared to CCL21 (*Figure 3B–C*). In addition, the recruitment of Rap1GAP-SmBiT to LgBiT-FYVE was blunted by the $G_{i/o}$ inhibitor pertussis toxin (PTX), and was not observed in mock-transfected HEK293 (*Figure 3—figure supplement 1A–C*). These results indicate that internalized CCR7 promotes endosomal $G_{i/o}$ activation.

To further validate if CCR7 activates $G_{i/o}$ at endosomes, we measured the translocation of RlucII-miniGi protein to the early endosomal marker rGFP-Rab5a by EbBRET in response to the CCR7 challenge (*Figure 3D*). Using this approach, CCL19 stimulation led to a strong and significant increase in EbBRET signaling between RlucII-miniGi and rGFP-Rab5a indicating that CCL19-bound CCR7 indeed activates $G_{i/o}$ robustly from endosomes (*Figure 3E*). In contrast, CCL21 stimulation provoked a small yet significant increase in EbBRET response, demonstrating that $G_{i/o}$ protein is activated from endosomes by CCL21-bound CCR7 to a minor extent (*Figure 3E*).

Similar trend was also observed by confocal microscopy visualization of CCR7-expressing HEK293 cells co-expressing Halo-miniGi and either the plasma membrane marker RFP-Lck or the early endosomal marker RFP-EEA1. In cells expressing RFP-Lck, CCR7 activation with either CCL19 or CCL21 leads to the recruitment of Halo-miniGi to the cell surface (*Figure 3F–G*). As expected, no significant difference between CCL19- and CCL21-stimulated translocation of Halo-miniGi to the plasma membrane was detected. However, in cells expressing RFP-EEA1, activation by CCL19, and to a minor degree CCL21, resulted in co-localization of Halo-miniGi and RFP-EEA1 at endosomes (*Figure 3H–I*). Together these results suggest that CCL19-CCR7 promotes robust G protein activation from endosomes whereas CCL21-CCR7 predominantly activates $G_{i/o}$ at the cell surface and to a modest degree from endosomes.

## CCR7 assembles distinct signalosomes upon stimulation with either CCL19 or CCL21

Limited knowledge exists regarding functional outcomes of endosomal G protein signaling by internalized GPCRs, particularly chemokine receptors. Thus, to examine the impact of compartmentalized CCR7 signaling on downstream effectors, we performed unbiased mass spectrometry (MS)-based proteome profiling using APEX2. APEX2 is an engineered ascorbate peroxidase enzyme that we fused to the CCR7 carboxy-terminal and uses $H_2O_2$ as an oxidant to catalyze a one-electron oxidation of various small molecule substrates including biotin-tyramide (*Figure 4A*). Oxidation of biotin-tyramide leads to the generation of a highly reactive and short-lived (<1 ms) biotin-phenoxyl radical that conjugates to endogenous proteins that are in close proximity to the APEX2 enzyme (~20 nm, *Figure 4A*; *Tan et al., 2020*). Thus, the APEX2 application allows for spatiotemporal control of the biotinylation process with high precision. The resulting biotinylated proteins are then enriched with pull-down experiments using neutravidin beads, and their identities are analyzed by MS (*Figure 4A*).

The CCR7-APEX2 construct expressed well at the cell surface (*Figure 4—figure supplement 1A*) and receptor functionality was confirmed by assessing its ability to promote βarr1 translocation to the plasma membrane and early endosomes in response to CCL19 challenge (*Figure 4—figure*

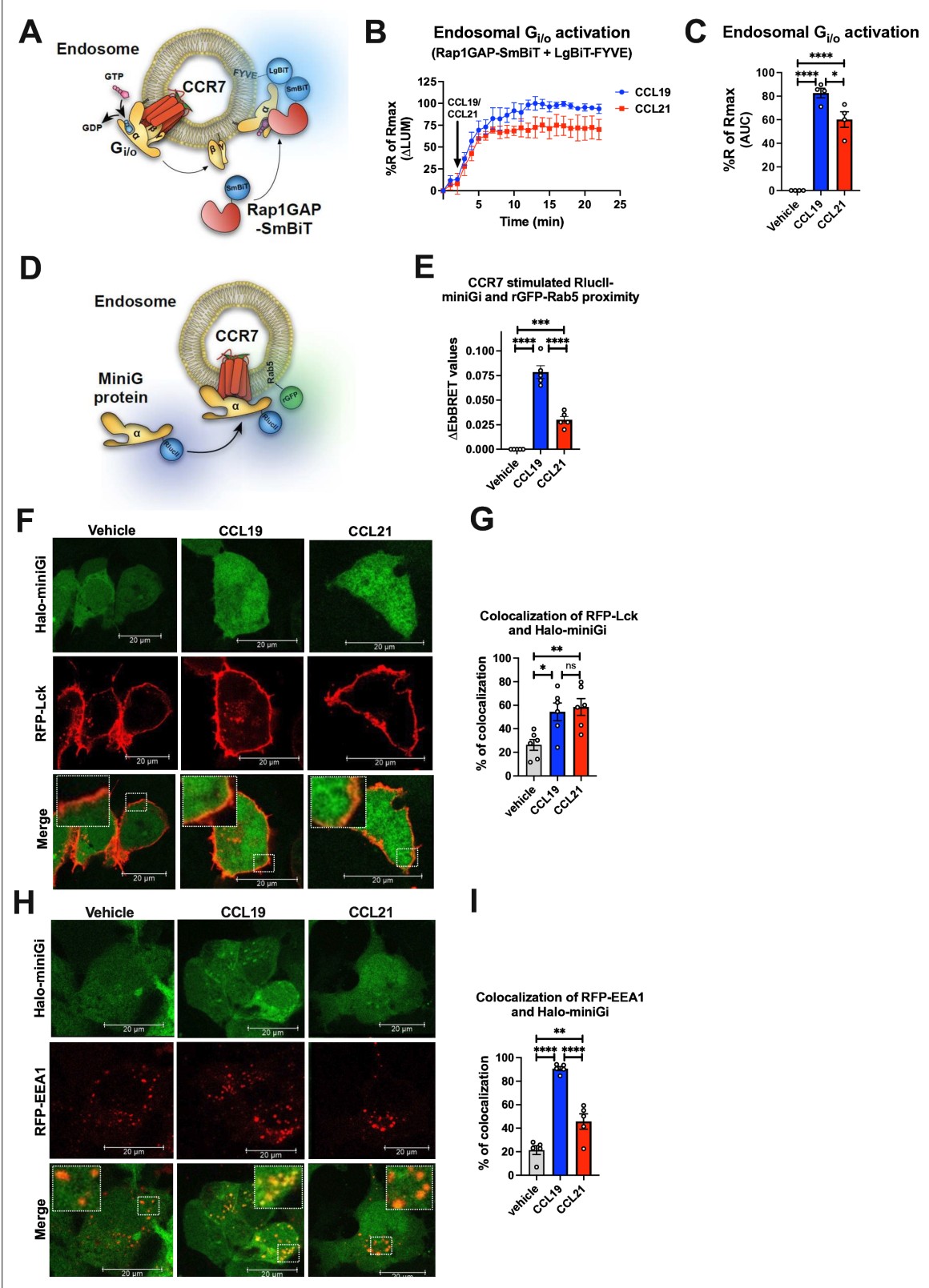

**Figure 3.** Endosomal $G_{i/o}$ activation by internalized CCR7. (**A**) Schematic representation of the experimental design used to measure luminescence upon proximity between Rap1GAP-SmBiT and LgBiT-FYVE in response to CCR7 activation. (**B**) Change in luminescence measured upon stimulation of HEK293-CCR7 cells expressing Rap1GAP-SmBiT and LgBiT-FYVE in response to 100 nM CCL19 or 100 nM CCL21 stimulation. The response to chemokine stimulation was normalized to vehicle control. (**C**) Area under the curve (AUC) was used to calculate the total response for each chemokine

*Figure 3 continued on next page*

*Figure 3 continued*

ligand. Data represent the mean ± SE of N=4 experiments. (**D**) Schematic representation of the enhanced bystander bioluminescence resonance energy transfer (EbBRET)-based assay to monitor proximity between RlucII-miniGi and the endosomal marker rGFP-Rab5 upon CCR7 activation from endosomes. (**E**) EbBRET measurements from CCR7-expressing HEK293 cells co-transfected with RlucII-miniGi and rGFP-Rab5 upon stimulation with 100 nM CCL19, 100 nM CCL21, or vehicle control. Data represents the mean ± SE from N=5 independent experiments. (**F**) Confocal microscopy imaging displaying CCR7-expressing HEK293 cells co-transfected with the plasma membrane marker RFP-Lck and Halo-miniGi. The cells were stimulated with 100 nM CCL19, 100 nM CCL21, or vehicle control for 10 min. Representative images from N=3 experiments. (**G**) Co-localization quantification analysis of RFP-Lck and Halo-miniGi from six representative confocal microscopy images per condition. (**H**) Confocal microscopy imaging displaying CCR7-expressing HEK293 cells co-transfected with the endosomal marker RFP-EEA1 and Halo-miniGi. The cells were stimulated with 100 nM CCL19, 100 nM CCL21, or vehicle control for 30 min. Representative images from N=3 experiments. (**I**) Co-localization quantification analysis of RFP-EEA1 and Halo-miniGi from five confocal microscopy images per condition. (**C, E, G, and I**) One-way ANOVA with Tukey's multiple comparison post hoc tests were applied to determine statistical differences between the distinct treatments (*$p<0.05$; **$p<0.01$; ***$p<0.001$; ****$p<0.0001$).

The online version of this article includes the following figure supplement(s) for figure 3:

**Figure supplement 1.** Control experiments to validate that the Rap1GAP-SmBiT/LgBiT-FYVE biosensor pair detects $G_{i/o}$ activation at endosomes upon chemokine stimulation and that this signal only is generated in CCR7-expressing cells.

---

*supplement 1B*). Additionally, CCL19 or CCL21 stimulation inhibited forskolin-induced cAMP production equally as expected (*Figure 4—figure supplement 1C*). In addition, we verified that the APEX2 portion of fusion proteins only biotinylated proteins in the presence of both biotin-tyramide and hydrogen peroxide as expected (*Figure 4B*). Using neutravidin beads, we demonstrated that proteins were pulled-down from HEK293-CCR7-APEX2 cell lysates where the biotinylation process had been initiated, whereas only negligible proteins were pulled-down in untreated cells (*Figure 4C*). Together these results indicate that both receptor and APEX2's enzyme functions of the CCR7-APEX2 fusion are intact.

Using our experimental setup, we stimulated HEK293-CCR7-APEX2 cells pretreated with biotin-tyramide with either CCL19 or CCL21 for 0 min, 2 min, 10 min, or 25 min. Hereafter, the biotinylation reaction was initiated by the addition of hydrogen peroxide for exactly 1 min, followed by extensive washing, cell lysis, enrichment of biotinylated proteins, and MS profiling.

From the MS profiling, we identified a total of 584 proteins, whose enrichment was changed significantly upon stimulation with either CCL19 or CCL21 ($p<0.05$ and change in intensity of $\log_2>1$) (*Figure 5—figure supplement 1A–C*). Among robustly enriched proteins by CCR7 activation, several are involved in functions related to vesicle trafficking, signal transduction, cytoskeletal dynamics, and cell migration among others (*Figure 5A*). Furthermore, as CCR7 is trafficked from the plasma membrane to endosomes within the first minutes of stimulation, we also included functionally validated APEX2-fused spatial references, PM-APEX2, ENDO-APEX2, and CYTO-APEX2 to control for the potential increase in random collisions between CCR7 and endosomal proteins that might occur upon receptor internalization (*Figure 5B-D*, *Figure 5—figure supplement 2*; *Lobingier et al., 2017*). Proteins enriched by both ENDO-APEX2 (as compared to both CYTO-APEX2 and PM-APEX2) and chemokine-stimulated CCR7-APEX2 include SNX3, RAB9A, ARH, and CCD93 (*Figure 5A and D*). Thus, these proteins might be enriched by CCR7-APEX2 due to chemokine-stimulated translocation from the plasma membrane to endosomes rather than participating in multiprotein functional complexes or signalosomes forming at CCR7.

As expected, proteins involved in receptor and vesicular trafficking including RIC1, SNX17, VA0D1, ARRB2, and DYN3, among others, were enriched over the entire time course of CCR7 activation (*Figure 5A*). Most of these proteins were more enriched by CCL19 stimulation as compared to the CCL21 challenge. In addition, we observed enrichment of endosomal proteins such as STX7, STX12, SNX3, and VTI1B for both CCL19 and CCL21 stimulation (*Figure 5A*). Interestingly, proteins involved in lysosomal trafficking including LAMP1, VPS16, VAMP7, WDR91, and PP4P1 were enriched by CCL19 and to a lesser degree by CCL21 after 25 min of stimulation (*Figure 5A*). Proteins involved in receptor trafficking via recycling endosomes or the trans-Golgi network such as SNX6, RAB7L, and GGA3 were also enriched at the later stages of CCR7 activation particularly when stimulated with CCL21 (*Figure 5A*). In addition to these, proteins associated with the Golgi network (SNX3, COG5, YIF1A, SC22B, and AP3S1) were enriched by either CCL19 or CCL21 suggesting general CCR7 trafficking to this subcellular compartment as previously reported (*Laufer et al., 2019*). These results

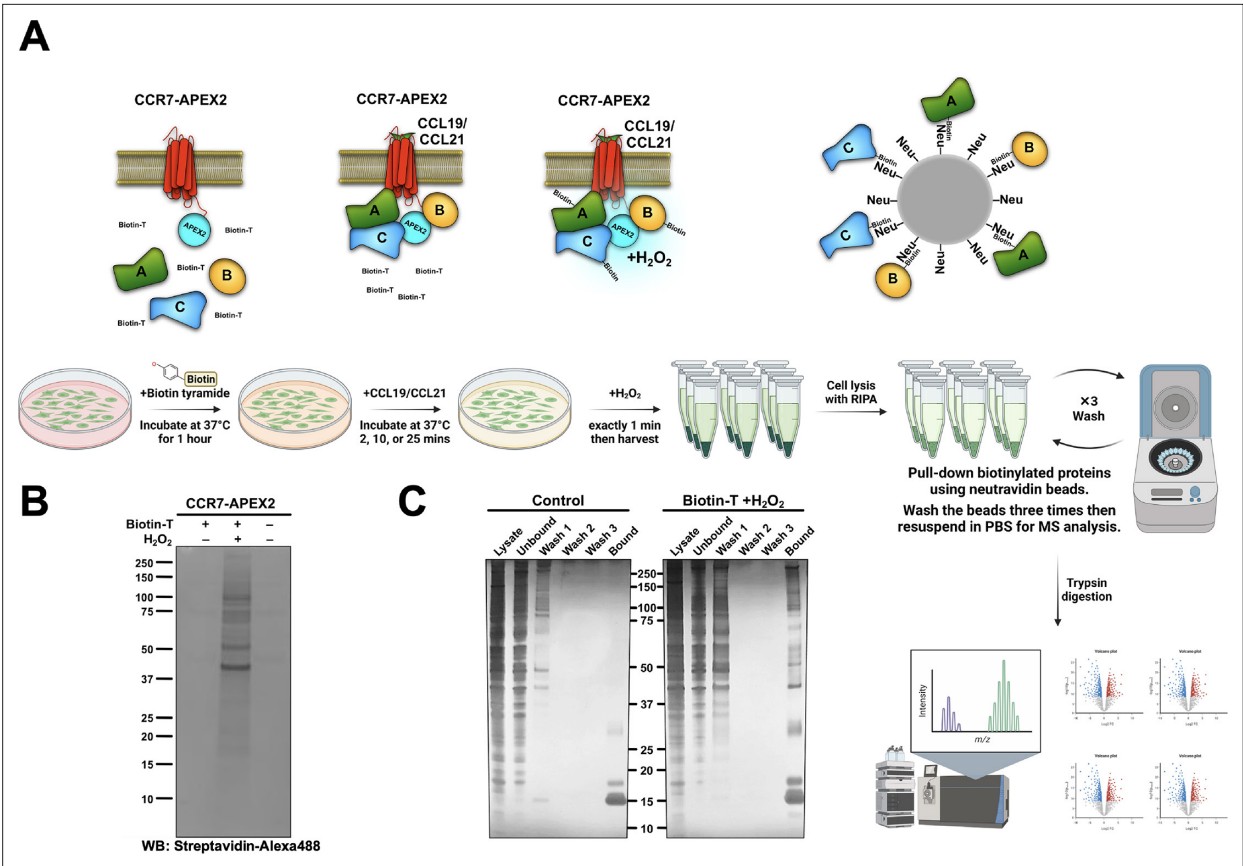

**Figure 4.** APEX2-mediated biotinylation of proteins in proximity of CCR7. (**A**) Schematic illustration of the workflow behind CCR7-APEX2-mediated biotinylation, created with BioRender.com. First, the cells are loaded with biotin-tyramide (biotin-T) followed by stimulation of CCR7 by 100 nM CCL19 or CCL21 for 0 min, 2 min, 10 min, or 25 min. For the last 1 min of chemokine-stimulation hydrogen peroxide is added, which initiates the APEX2-mediated oxidation of biotin-tyramide into highly reactive and short-lived radicals. These radicals bind to proteins within close proximity to the APEX2 enzyme (~20 nm), and thus, label proteins that are in complex with or in close proximity chemokine-stimulated CCR7. Next, the cells are lysed and the resulting biotinylated proteins are captured on neutravidin (Neu) beads followed by extensive washing. Finally, all biotin-labeled proteins are eluted, identified, and analyzed by LC-MS. (**B**) Western blot analysis of HEK293-CCR7-APEX2 cell lysates, which shows that the biotinylation only takes place in the presence of both biotin-tyramide and hydrogen peroxide. Biotinylated proteins were detected using streptavidin-alexa488. (**C**) Silver staining of the pull-down experiments demonstrating that the enrichment of biotinylated proteins using neutravidin-coated beads is highly specific. In the control sample, the bound fraction only displays neutravidin band on the SDS-PAGE, whereas the labeled sample shows multiples bands of biotinylated proteins.

The online version of this article includes the following source data and figure supplement(s) for figure 4:

**Source data 1.** PDF file containing original western blot for *Figure 4B* and silver stain gel for *Figure 4C*, indicating the relevant bands and treatments.

**Source data 2.** Original files for western blot and silver stain analysis displayed in *Figure 4C*.

**Figure supplement 1.** Surface expression and validation of functionality of the CCR7-APEX2 construct.

indicate that the trafficking pattern of CCR7 upon chemokine stimulation is tightly regulated between lysosomal degradation and recycling.

Among enriched signaling proteins, we observed robust enrichment of proteins regulating inositol 1,4,5-triphosphate signaling (ERLN1/2, IP3KA), diacylglycerol signaling (DGKQ), calcium mobilization at the endoplasmic reticulum (TMCO1), and activity of the Rho GTPases (*Figure 5A and E–G*). These are all involved in signaling events downstream of heterotrimeric G protein activation. Another interesting signaling protein enriched by CCR7 activation includes a member of the eyes absent (EYA) subfamily of protein, EYA2. This protein has been shown to interact with and regulate $G\alpha_{i/o}$ and $G\alpha_z$ subunits (*Embry et al., 2004*; *Fan et al., 2000*), and EYA4, a member of the same family, was recently identified in a similar APEX2-based study of the μ-opioid receptor (*Polacco et al., 2024*). We also detected three novel interaction partners GRIP2, MARK4, and EI24 as being among the highest levels regardless of the activation ligand or time (*Figure 5A*, *Figure 5—figure supplement 1B*).

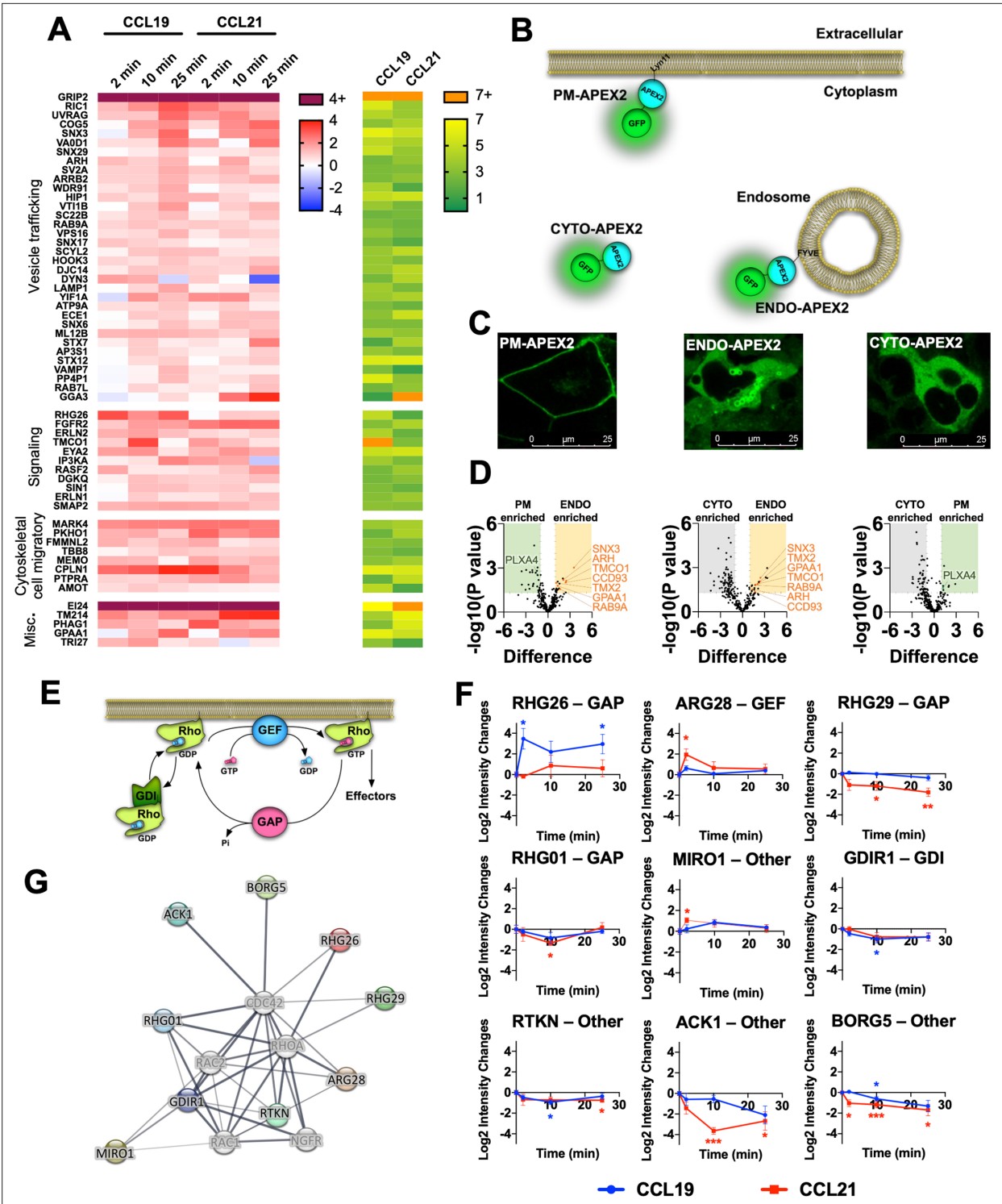

**Figure 5.** Identification of protein enrichment in the proximal proteome of CCR7 following agonist stimulation. (**A**) Heatmap visualizing selected proteins with significant increase (p<0.05 and Log$_2$ fold-increase >1) in the proximal proteome of CCR7 for at least one timepoint following chemokine stimulation. Data represent the mean of N=4 experiments. The proteins are clustered according to their function with corresponding significance score, which was calculated by combining the absolute value of Log$_2$ fold-change and -Log10 p value as previously suggested (**Xiao et al., 2014**). (**B**) Schematic representation showing organelle markers fused to APEX2 used for spatial controls. (**C**) Confocal microscope images of HEK293-CCR7 cells expressing each of the organelle markers (PM-APEX2, ENDO-APEX2, and CYTO-APEX2) at their respective subcellular locations. (**D**) Volcano plots of enrichment differences of PM vs CYTO, ENDO vs PM, and ENDO vs CYTO, respectively. Proteins represented as green dots are significant in PM-CYTO/ENDO-PM pair for plasma membrane protein, and red dots in ENDO-CYTO/ENDO-PM pair for endosomal proteins. (**E**) Schematic illustration

*Figure 5 continued on next page*

*Figure 5 continued*

showing the roles of RhoGAP, RhoGEF, and RhoGDI in regulation of the family of Rho-GTPase function. (**F**) Differential enrichment of RhoGAP, RhoGEF, RhoGDI, and other proteins that regulate Rho-GTPase signaling with significant change in the CCR7 proximal proteome following chemokine stimulation. Student's *t*-tests were applied to determine statistical differences between unstimulated cells and chemokine stimulation at different time points (\*p<0.05; \*\*p<0.01; \*\*\*p<0.001) (**G**) Interaction network of RhoGAP, RhoGEF, and RhoGDI proteins with significant change in the CCR7 proximal proteome following chemokine stimulation.

The online version of this article includes the following source data and figure supplement(s) for figure 5:

**Figure supplement 1.** Identification of enrichment in proximal proteome of CCR7 following agonist stimulation.

**Figure supplement 2.** Western blot analysis of HEK293 cells transfected with PM-APEX2, ENDO-APEX2, or CYTO-APEX2 shows that the biotinylation only takes place in the presence of both biotin-tyramide and hydrogen peroxide, and that they have different proximity proteome.

**Figure supplement 2—source data 1.** PDF file containing original western blot, indicating the relevant bands and treatments.

**Figure supplement 2—source data 2.** Original file for the western blot.

## Endosomal $G_{i/o}$ signaling is important for Rac1 activation and chemotaxis

Rho GTPases such as RhoA, Rac1, Cdc42 regulate important aspects of chemotactic cell migration (*Schiefermeier et al., 2011*). Activity of these effectors are regulated by other proteins such as Rho GTPase-activating proteins (GAPs), Rho guanine nucleotide exchange factors (GEFs), and Rho GDP dissociation inhibitors (GDIs) (*Figure 5E*). Our APEX2-based proteomics results demonstrate that a number of these Rho GTPase regulators are enriched by chemokine-stimulation of CCR7, and thus, potentially form part of larger CCR7 signalosomes that control downstream Rho GTPase activity (*Figure 5F–G*). Interestingly, some of these regulators, including RHG26, ARG28, RHG29, ACK1, and BORG5, were differentially enriched upon CCL19 or CCL21 stimulation (*Figure 5F*). Therefore, it is possible that CCR7 signaling modulates the activity of cell migratory Rho GTPases specifically at distinct cellular locations. To test this hypothesis, we focused on RhoA, Rac1, and Cdc42. Upon activation of RhoA, Rac1, and Cdc42, protein kinase N1 (PKN1), p21 activated kinase 1 (PAK1), and Wiskott-Aldrich syndrome protein 1 (WAS1), are respectively recruited. We assessed the proximity of these biosensor pairs (*Leng et al., 2013*; *Inoue et al., 2019*) in real-time upon chemokine stimulation by the NanoBiT approach in HEK293-CCR7 cells (*Figure 6A*). Using this setup, we found that both CCL19 and CCL21 stimulated RhoA and Cdc42 activation to a similar extent, which indicates that the spatial aspect of G protein activation does not appear to influence RhoA and Cdc42 signaling (*Figure 6B–C, and F–G*). In contrast, CCL19 promoted Rac1 signaling to a significantly greater extent than CCL21 stimulation (*Figure 6D–E*). No change in RhoA, Rac1, or Cdc42 activity upon CCL19/CCL21 challenge was detected HEK293 cells that do not express CCR7 (*Figure 6—figure supplement 1A–C*). However, CXCL12 stimulation led to enhanced activity of RhoA, Rac1, or Cdc42 in HEK293 cells, which express CXCR4 (*Figure 6—figure supplement 1D–F*). Pre-incubating the HEK293-CCR7 cells with the $G_{i/o}$ inhibitor PTX, virtually eliminated the chemokine-induced activation of Rac1, indicating that CCR7-stimulateted Rac1 signaling is mediated via a $G_{i/o}$-dependent mechanism (*Figure 6H–I*). Interestingly, pre-treatment of the HEK293-CCR7 cells with the endocytosis inhibitor Dyngo-4a (*Jensen et al., 2017*) reduced chemokine-stimulated CCR7 internalization and Rac1 activation robustly (*Figure 6J-K*, *Figure 6—figure supplement 1G-H*). This reduction was significantly smaller as compared to cells pre-incubated with an inactive Dyngo compound (*Jensen et al., 2017*; *Figure 6J-K*, *Figure 6—figure supplement 1G-H*). Furthermore, overexpression of the hemagglutinin-tagged dominant negative dynamin 1 K44A mutant (HA-Dyn-K44A) diminished chemokine-activated CCR7 internalization and Rac1 activity suggesting that CCR7-mediated Rac1 activation depends on receptor internalization (*Figure 6L-M*, *Figure 6—figure supplement 1I-K*). Also, pre-incubating the cells with the endocytosis inhibitor PitStop2, which acts on clathrin rather than dynamin, led to a severe reduction in chemokine-stimulated CCR7 internalization and Rac1 activity (*Figure 6N-O*, *Figure 6—figure supplement 1L-M*). Finally, we mutated all potential serine/threonine phosphorylation sites in the carboxy-terminal tail of CCR7 and found that this CCR7-ΔST construct expresses well at the cell surface, but does not internalize upon CCL19 stimulation (*Figure 6—figure supplement 1N-P*). When expressed in HEK293 cells, the ability of CCR7-ΔST to activate Rac1 in response to chemokine-stimulation is severely reduced as compared to wild-type CCR7. These results suggest that signaling by internalized CCR7 is a mechanism to activate Rac1.

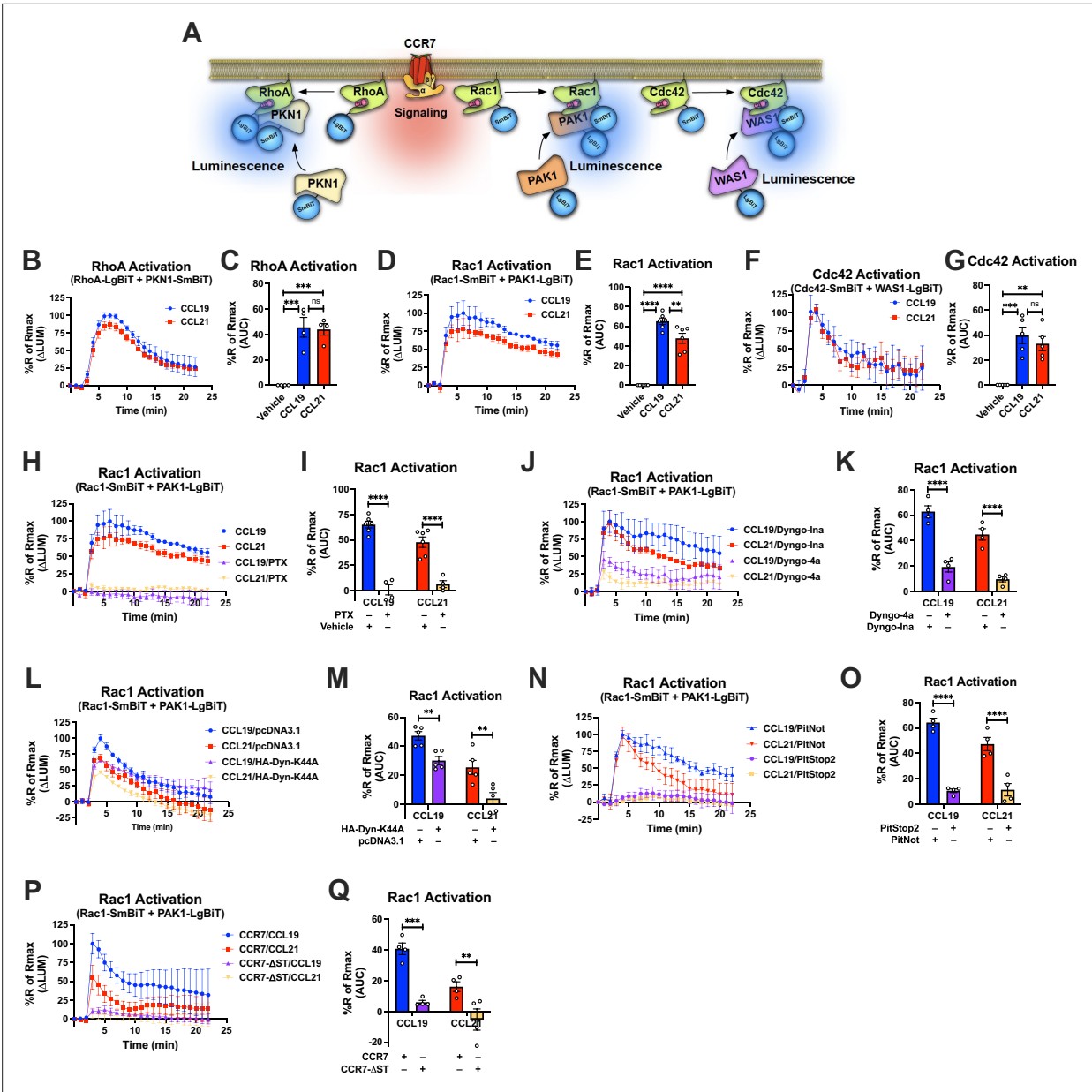

**Figure 6.** Compartmentalized CCR7 signaling and regulation of RhoA, Rac1, and Cdc42 signaling as well as chemotaxis. (**A**) Schematic description of the RhoA/Rac1/Cdc42 NanoBiT assay. (**B, D**, and **F**) Change in luminescence measured upon stimulation of HEK293-CCR7 cells expressing (**B**) SmBiT-PKN1/LgBiT-RhoA, (**D**) SmBiT-Rac1/LgBiT-PAK1, or (**F**) SmBiT-Cdc42/LgBiT-WAS1 in response to 100 nM CCL19 or 100 nM CCL21 stimulation. The response to chemokine stimulation was normalized to vehicle control. (**C, E**, and **G**) Area under the curve (AUC) was used to calculate the total (**C**) RhoA, (**E**) Rac1, and (**G**) Cdc42 activity response for each chemokine ligand. Data represent the mean ± SE of N=4–6 experiments. (**H, J, L**, and **N**) HEK293-CCR7 cells were either pre-treated with (**H**) 100 ng/ml PTX or control buffer for 16 hr, (**J**) 30 µM of the endocytosis inhibitor Dyngo-4a or the inactive Dyngo control compound for 30 min, (**L**) co-transfected with the dominant negative HA-Dyn-K44A mutant or mock pcDNA3.1 control plasmid, or (**N**) pre-treated with 10 µM of the endocytosis inhibitor PitStop2 or the inactive PitNot control compound for 30 min. (**P**) HEK293 cells were co-transfected with either wild-type CCR7 or CCR7-ΔST. (**H, J, L, N**, and **P**) Changes in luminescence were measured upon stimulation of HEK293-CCR7 cells expressing SmBiT-Rac1/LgBiT-PAK1 in response to 100 nM CCL19 or 100 nM CCL21 stimulation. The response to chemokine stimulation was normalized to vehicle control. (**I, K, M, O**, and **Q**) AUC was used to calculate the total Rac1 activity response for each condition. Data represent the mean ± SE of N=4–6 experiments. (**C, E, and G**) One-way ANOVA or (**I, K, M, O, and Q**) two-way ANOVA with (**C, E, and G**) Tukey's or (**I, K, M, O, and Q**) Sidak's multiple comparison post hoc tests were performed to determine statistical differences between the distinct treatments (*p<0.05; **p<0.01; ***p<0.001; ****p<0.0001).

The online version of this article includes the following source data and figure supplement(s) for figure 6:

*Figure 6 continued on next page*

*Figure 6 continued*

**Figure supplement 1.** Control experiments to validate that the RhoA, Rac1, and Cdc42 biosensors detect activity in response to chemokine receptor stimulation and that experimental manipulations to block receptor internalization are effective.

**Figure supplement 1—source data 1.** PDF file containing original western blots for *Figure 6—figure supplement 1K*, indicating the relevant bands and treatments.

**Figure supplement 1—source data 2.** Original files for the western blot analysis displayed in *Figure 6—figure supplement 1K*.

As Rac1 is a key regulator of chemotaxis, we next tested the ability of CCL19 and CCL21 to promote cell migration of endogenously CCR7-expressing Jurkat T cells (*Yang et al., 2011*) towards either CCL19 or CCL21. As in HEK293-CCR7 cells, CCL19, and CCL21 stimulation led to similar inhibition of forskolin-stimulated cAMP production in Jurkat T cells confirming that CCR7 activation by either chemokine results in comparable stimulation of $G_{i/o}$ signaling (*Figure 7A*). Interestingly, Jurkat T cells migrated more efficiently towards a gradient of CCL19 as compared with CCL21, which was assessed by the transwell chemotaxis assay (*Figure 7B*). These findings raised the possibility that endosomal $G_{i/o}$ signaling plays a major role in chemotaxis. To test this hypothesis, we pre-treated the Jurkat T cells with the $G_{i/o}$ inhibitor PTX, which almost completely abolished the ability of the cells to migrate towards both CCL19 and CCL21 (*Figure 7C*). Interestingly, pre-incubation with the endocytosis inhibitors Dyngo-4a or PitStop2 led to a significant reduction in chemotaxis of both CCL19 and CCL21-stimulated cells as compared to the inactive Dyngo or PitNot compounds, respectively (*Figure 7D–E*).

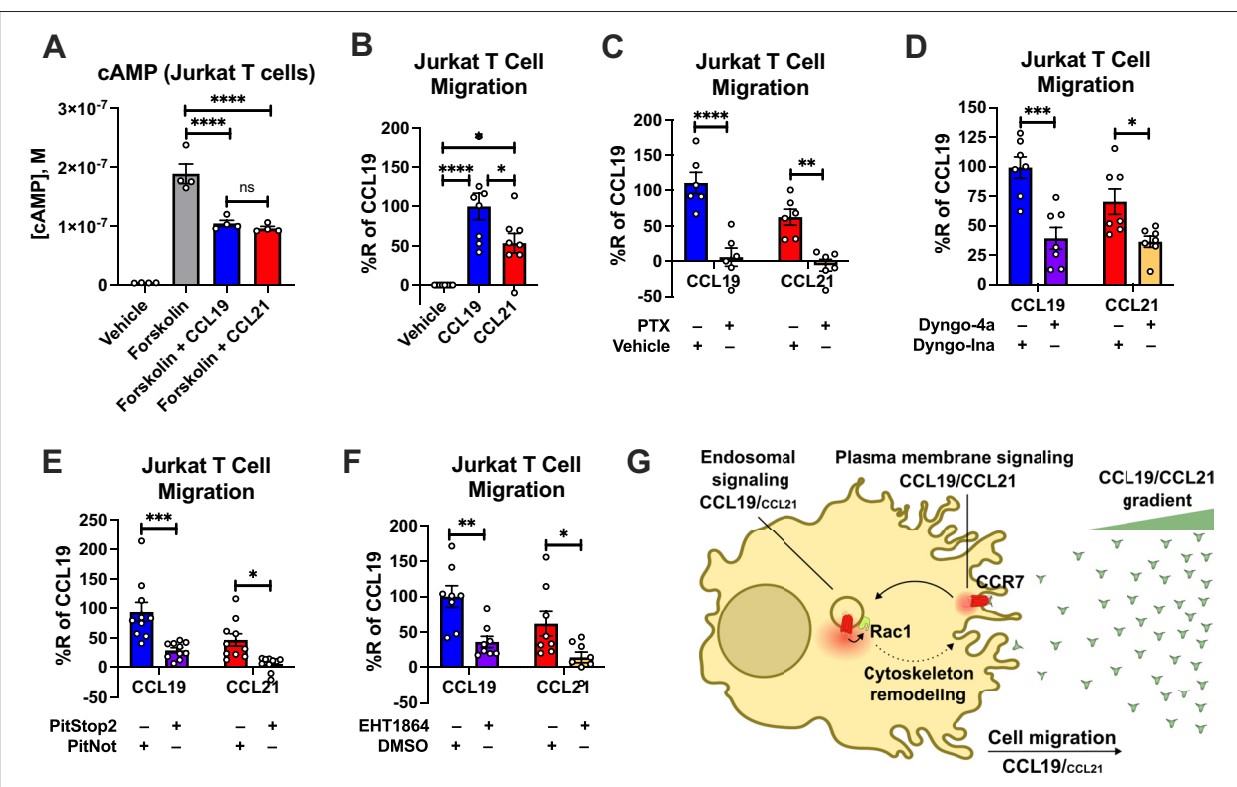

**Figure 7.** CCR7-mediated chemotaxis of Jurkat T cells. (**A**) CCR7 stimulation of $G_{i/o}$ signaling in Jurkat T cells. Jurkat T cells were challenged with 10 µM forskolin (or vehicle buffer) to increase cAMP production either with or without 100 nM CCL19 and 100 nM CCL21. The accumulation of cAMP was determined using the Cisbio cAMP dynamic assay. Data represent the mean ± SE of N=4 experiments. (**B**) Chemotaxis of Jurkat T cells towards a gradient of 100 nM CCL19, 100 nM CCL21, or vehicle control. Total cells migrated through the transwell chamber were normalized to the CCL19 response. Data represent the mean ± SE of N=8 experiments. (**C–F**) Chemotaxis of Jurkat T cells that were either pre-treated with (**C**) 100 ng/ml PTX or control buffer for 16 hr, (**D**) 30 µM of the endocytosis inhibitor Dyngo-4a or the inactive Dyngo control compound for 30 min, (**E**) 10 µM of the endocytosis inhibitor PitStop2 or the inactive PitNot control compound for 30 min, (**F**) or 10 µM of the Rac1 inhibitor EHT1864 or DMSO-containing control buffer for 30 min. Data represent the mean ± SE of N=6–10 experiments. (**A–B**) One-way ANOVA or (**C–F**) two-way ANOVA with (**A and C–F**) Sidak's or (**B**) Tukey's multiple comparison post hoc tests were performed to determine statistical differences between the distinct treatments (*p<0.05; **p<0.01; ***p<0.001; ****p<0.0001). (**G**) Schematic illustration of how endosomal CCR7 signaling promotes chemotaxis.

Finally, as CCR7-mediated endosomal $G_{i/o}$ signaling was shown to enhance Rac1 activity, we pre-treated Jurkat T-cells with the Rac1 inhibitor EHT1864. As expected, EHT1864 reduced Jurkat T-cell chemotaxis towards both CCL19 and CCL21 (*Figure 7F*). Collectively, these findings suggest that endosomal $G_{i/o}$ signaling by internalized CCR7 promotes chemotaxis of Jurkat T cells.

## Discussion

Historically, GPCRs have been thought to activate heterotrimeric G proteins exclusively from the cell surface, which leads to downstream signaling events that regulate cell physiological responses. This G protein signaling has been characterized as short-lived due to a specialized desensitization mechanism that includes receptor phosphorylation by GPCR kinases and subsequent recruitment of βarrs to the phosphorylated receptor. The GPCR–βarr interaction both uncouples G protein from the receptor and promotes receptor endocytosis, which are hallmarks of receptor desensitization.

This paradigm, however, has been challenged by multiple observations, which show GPCRs that continue to activate G proteins from internalized compartments, such as from endosomes. Paradoxically, GPCRs that promote endosomal G protein signaling most substantially include receptors such as the $V_2R$, parathyroid hormone receptor 1, protease-activated receptor type 2, and neurokinin 1 receptor that all associate strongly with βarrs via phosphorylation site clusters located at their carboxy-terminal tail (*Jensen et al., 2017*; *Thomsen et al., 2016*; *Wehbi et al., 2013*; *Feinstein et al., 2013*; *Jimenez-Vargas et al., 2018*). Recently, we demonstrated that GPCRs with these carboxy-terminal clusters associate with βarrs exclusively through this region to form tail conformation GPCR–βarr complexes (*Cahill et al., 2017*). Since βarrs do not block the G protein-binding site in this tail conformation, the receptor can associate with βarrs and G proteins simultaneously to form GPCR–G protein–βarr megaplexes (*Thomsen et al., 2016*; *Nguyen et al., 2019*). The assembly of these megaplexes allows the receptor to continue to stimulate G protein signaling while being internalized into endosomes by βarrs.

The chemokine receptor CCR7 contains serine/threonine phosphorylation site clusters on its carboxy-terminal tail (*Table 1*), which have been shown to be fully phosphorylated upon CCL19 stimulation, but not CCL21 stimulation (*Kohout et al., 2004*; *Zidar et al., 2009*). In the present study, we also found a correlation between this reported CCR7 phosphorylation pattern, formation of tail conformation CCR7–βarr1 complexes, simultaneous recruitment of G protein and βarr to CCR7, robust endosomal G protein signaling, Rac1 signaling, and cell migration (*Figures 1–3, 6D-E, H-Q and 7A-F*). As most chemokine receptors also contain serine/threonine clusters in their carboxy-terminal tails similar to CCR7, it not only raises the possibility that these receptors can promote endosomal G protein signaling, but also that this mode of signaling regulates important aspects of the physiological functions associated with these receptors (*Table 1*). In fact, removing the carboxy-terminal tail or mutating its phosphorylation sites in the chemokine receptors CXCR1-4 reduces their ability to internalize and promote cellular chemotaxis towards a chemokine-gradient (*Roland et al., 2003*; *Richardson et al., 2003*; *Colvin et al., 2004*). Similar reduction in Akt signaling and chemotaxis of immune cells towards gradients of different chemokines were observed upon pharmacological inhibition of endocytosis (*Jacques et al., 2015*; *English et al., 2018*). Additionally, removal of intracellular loops in CXCR4 reduces G protein activation and is associated with decreased capacity of CXCR4 to promote cell migration indicating a co-dependency of G protein signaling and receptor internalization on this cell physiological function (*Roland et al., 2003*). Notably, the chemokine receptor CCR1 displays constitutive G protein activation and internalization in HEK293 cells by a mechanism where $G\alpha_{i/o}$ and βarr complex with the receptor simultaneously (*Gilliland et al., 2013*). This constitutive CCR1-mediated G protein activation from internalized compartment was reported to stimulate cell migration. Finally, sustained $G_{i/o}$ signaling by internalized sphingosine-1-phophate type 1 receptor has been shown to promote cell migration in human umbilical vein endothelial cells (*Mullershausen et al., 2009*). Thus, previous observations together with our results are supportive of certain aspects of cell migration being regulated by internalized chemokine receptor signaling.

To obtain a greater understanding of the impact of endosomal chemokine receptor signaling on cell physiological responses, we applied an unbiased MS-based proteome profiling approach using CCR7-APEX2 expressing HEK293 cells. We found that CCR7 activation leads to enrichment of multiple proteins of key functions including vesicle trafficking, signal transduction, and cytoskeletal dynamics/cell migratory, among others (*Figure 5A*). The proteins GRIP2 and EI24 were among the

most enriched following CCR7 activation. GRIP2 and EI24 have not previously been reported as having roles in GPCR biology. GRIP2 has been shown to bind AMPA ionotropic glutamate receptors via PDZ-domains to regulate their intracellular trafficking (*Mao et al., 2010*; *Dong et al., 1999*). Whether GRIP2 plays similar role in the trafficking of chemokine receptors remains to be tested. EI24 is a protein whose expression is enhanced by p53 activation, and plays a role in growth suppression and apoptosis as well as in autophagy through formation of degradative autolysosomes (*Tian et al., 2010*). As with GRIP2, the potential role of EI24 in chemokine receptor biology is unknown. Another highly enriched protein was cytosolic EYA2, which interacts with members of the *Sine oculis* (Six) family of homeodomain transcription factors. This interaction facilitates the translocation of EYA proteins into the nucleus, where the EYA/Six complex regulates transcription of a number of genes involved in development and cancer (*Tadjuidje and Hegde, 2013*). Interestingly, EYA2 can interact directly with $G\alpha_{i/o}$ and $G\alpha z$ subunits and regulate their activity (*Embry et al., 2004*; *Fan et al., 2000*). This interaction prevents their translocation to the nucleus, and thus, block their role as transcription co-factors (*Embry et al., 2004*). Coincidently, EYA4 was recently identified in a similar APEX2-based study of the μ-opioid receptor, and thus, might be more involved in downstream signaling responses of $G_{i/o}$-coupled receptors such as chemokine receptors than currently appreciated (*Polacco et al., 2024*).

Another protein that was robustly enriched upon CCR7-APEX2 activation was the Rho GTPase GAP RHG26. Notably, this enrichment was exclusively observed when HEK293-CCR7-APEX2 cells were stimulated by CCL19, but much less by CCL21 (*Figure 5A and F*). Other regulators of Rho GTPases were also enriched by CCR7 activation and some of them were differentially enriched by either CCL19 or CCL21 stimulation (*Figure 5F*). These results raise the possibility that CCR7 assembles compartment-specific signalosomes that could play an important role in downstream Rho GTPase activity. In fact, we discovered that the Rho GTPase Rac1 was specifically activated by G protein signaling by internalized CCR7 whereas the Rho GTPases RhoA and Cdc42 were stimulated equally by plasma membrane and endosomal CCR7 signaling (*Figure 6*). Rac1 plays a key role in cell migration, which we further demonstrated in this study (*Figure 7F*), and can be activated in endosomes by RhoGEFs such as Tiam1 and Vav1 (*Palamidessi et al., 2008*; *Laufer et al., 2019*). Previously, a chemokine-stimulated association between CCR7 and Vav1 at endomembranes was reported, which depends on recruitment of βarrs and Src (*Laufer et al., 2019*). The formation of this signaling complex was shown to directly activate Rac1. In contrast to our findings, the formation of this signaling complex did not depend on $G_{i/o}$ protein and occurred mostly at the Golgi network (*Laufer et al., 2019*). Interestingly, we also observed an enrichment of Golgi network proteins upon CCL19 and CCL21 stimulation of the CCR7-APEX2 construct (*Figure 5A*), which raises the possibility that Rac1 can be activated by distinct CCR7-mediated mechanisms at different subcellular compartments.

Activation of Rac1 leads to its translocation to the cell surface via recycling endosomes, a process reported to occur in response to stimulation of CCR7 as well as several receptor tyrosine kinases (*Figure 7G*; *Palamidessi et al., 2008*; *Laufer et al., 2019*). From this subcellular region, Rac1 activates the major effector PAK1, which phosphorylates LIM kinase and cortactin, among others, to coordinate actin polymerization at the plasma membrane region (*Tebar et al., 2018*). This polymerization leads to formation of cytoskeletal actin filaments that serve as underlying stabilizing structures of newly formed filopodia and lamellipodia (*Tebar et al., 2018*). As these membrane protrusions constitute a major mechanistic step during cell migration, endosomal Rac1 activation plays a central role in this cell physiological process (*Schiefermeier et al., 2011*). To this end, we found that CCL19 triggers chemotaxis of Jurkat T cells more robustly as compared to CCL21 despite both chemokines activating CCR7-mediated G protein signaling to the same extent (*Figure 7A–B*). Similar findings that demonstrate CCL19's superior ability to promote chemotaxis of other CCR7-expressing cells as compared to CCL21 have previously been reported (*Kohout et al., 2004*; *Hjortø et al., 2016*; *Ricart et al., 2011*; *Jørgensen et al., 2018*; *Bardi et al., 2001*). Furthermore, using pharmacological intervention we demonstrated that $G_{i/o}$ signaling from internalized compartments play a significant role in the chemotaxis response to CCR7 activation (*Figure 7C–E*). Thus, the findings from our study highlight endosomal G protein signaling by some chemokine receptors as a potential mechanism that regulates key aspects of cell migration.

# Materials and methods

## Reagents

Hank's balanced salt solution (HBSS), forskolin, G418, HEPES, NaCl, phosphate saline buffer (PBS) tablets, 3-isobutyl-1-methylxanthine (IBMX), KCl, MgCl$_2$, NaHCO$_3$, NaH$_2$PO$_4$, glucose, sodium ascorbate, sodium azide, biotin-tyamide, sodium deoxycholate, Tris-HCl, H$_2$O$_2$, sodium dodecyl sulfate (SDS), phenylmethylsulfonyl fluoride (PMSF), ethanol, chloroacetamide (CAA), Tween-20, tris(2-carboxyethyl)phosphine (TCEP), ethylenediaminetetraacetic acid (EDTA), CaCl$_2$, rabbit anti-β-tubulin antibody (cat. no. T2200), and anti-FLAG M2-HRP conjugated antibody (Cat. A8592, Sigma-Aldrich) were all purchased from Sigma-Aldrich. Dulbecco's Modified Eagle's Medium (DMEM) high glucose, Opti-MEM reduced serum/no phenol red, Dulbecco's PBS (DPBS), fetal bovine serum (FBS), penicillin/streptomycin, Lipofectamine 3000, zeocin, puromycin, donkey anti-mouse IgG antibody conjugated to Alexa Fluor 488 (cat. no. A21202), SuperSignal West Pico PLUS Chemiluminescent Substrate, and silver stain kit were all purchased from Thermo Fisher Scientific. Coelenterazine h, and coelenterazine 400 a were purchased from NanoLight Technology. NeutrAvidin agarose, paraformaldehyde, and Triton X-100 were purchased from Fisher Scientific. CCL19 was purchased from Chemotactics. CCL21 was purchased from GenScript. Trolox was purchased from Millipore, and Halo Tag ligand was purchased from Promega. Salmon sperm DNA was purchased from Invitrogen. Linear polyethyleneimine 25 K (PEI) was purchased from Polysciences. Mouse anti-HA antibody (cat. no. sc-7392) was purchased from Santa Cruz. Donkey anti-rabbit IgG F(ab')2 conjugated to horseradish peroxidase (HRP; cat. no. NA9340) was purchased from Cytiva.

## Plasmid constructs

Human CCR7 in pcDNA3 was previously described (*Zidar et al., 2009*). Human CCR7 was N-terminally tagged with an influenza hemagglutinin signal sequence followed by a FLAG-tag (MKTIIAL-SYIFCLVFA +DYKDDDDK) into pTWIST/CMV/Zeo and synthesized by Twist Bioscience. For CCR7 trafficking NanoBiT assays, a SmBiT (*Inoue et al., 2019*) was C-terminally tagged to the human CCR7 sequence through a linker (GGSG) and synthesized by Twist Bioscience. The CCR7-ΔST and CCR7-ΔST-SmBiT constructs were synthesized with 11 mutations of serine and threonine residues in the receptor C-terminal tail (S348A, S356A, S357A, S364A, S365A, S367A, T372A, T373A, T374A, T375A, and S377A). For βarr1 trafficking NanoBiT assays, a SmBiT was N-terminally tagged to the human βarr1 sequence through a flexible linker (GGSGGGGSGGSSSGG) and synthesized by Twist Bioscience. In addition, the plasma membrane-anchored LgBiT-CAAX and endosomally anchored LgBiT-FYVE have previously been described (*Latorre et al., 2022*). For proximity biotin labeling, CCR7 was N-terminally tagged with an influenza hemagglutinin signal sequence followed by a FLAG-tag, and with APEX2 added to the C-terminus through a flexible linker (GGSGGGGSGGSSSGG) and synthesized into pTwist/CMV/Puro using the EcoRI and NheI restriction sites of the vector by Twist Bioscience. Lyn11-GFP-APEX2 (PM-APEX), 2×FYVE-GFP-APEX2 (ENDO-APEX), and GFP-APEX2 (CYTO-APEX) were a kind gift from Dr. Mark von Zastrow and were designed as plasma membrane, endosomal, and cytosolic spatial references, respectively (*Lobingier et al., 2017*). The CAMYEL biosensor was previously described (*Jiang et al., 2007*) as well as RlucII-βarr1 and RlucII-βarr1-ΔFL (*Cahill et al., 2017*; *Quoyer et al., 2013*). The RlucII-βarr2 construct (*Renilla* Luciferase II in the carboxy-terminal of βarr2) was built by replacing the GFP10-EPAC sequence from the previously published GFP10-EPAC-RlucII (*Quoyer et al., 2013*; *Zimmerman et al., 2012*) with the coding sequence of human βarr2. The rGFP-CAAX and rGFP-Rab5a are cloned into pcDNA3.1(+) and were previously described (*Namkung et al., 2016*). MiniGi (*Nehmé et al., 2017*) (also referred to as miniGsi) and miniGs (*Nehmé et al., 2017*) were N-terminally tagged with a LgBiT (*Inoue et al., 2019*) through a linker (GGSG), and the LgBiT was N-terminally tagged with a nuclear export signal (MLQNELALKLAGLDINKT) via a linker (GGSG), and synthesized into pTwist/CMV using the NotI and BamHI restriction sites of the vector by Twist Bioscience. Rap1GAP (*Avet et al., 2022*) was C-terminally tagged with a SmBiT (*Inoue et al., 2019*) through a linker (GSAGTGGRAIDIKLPAT) and synthesized into pTwist/CMV using the NotI and BamHI restriction sites of the vector by Twist Bioscience. Human βarr1 was also N-terminally tagged with SmBiT (*Inoue et al., 2019*) through a linker (GGSG) synthesized by Twist Bioscience. The RlucII-miniGsi was synthesized by Twist Bioscience. The Venus tag from the NES-venus-mGsi contruct (*Wan et al., 2018*) was replaced by a *Renilla* Luciferase II and cloned in pTwistCMV expression vector using HindIII and NheI restriction sites of the vector. The NanoBiT-RhoA sensor comprising the SmBiT-RhoA

and the LgBiT-PKN1-GDB constructs was described previously (*Inoue et al., 2019*). The NanoBiT-Rac1 sensor comprising the SmBiT-Rac1 and the LgBiT-PAK1-GDB constructs and the NanoBiT-Cdc42 sensor comprising the SmBiT-Cdc42 and the LgBiT-WAS-GDB constructs were generated by replacing firefly luciferase fragments of previously described Rac1 and Cdc42 constructs (*Leng et al., 2013*) with the NanoBiT fragments. Specifically, the human Rac1 (residues 2–192) and the GTPase-binding domain (GBD) of the human PAK1 (residues 67–150) were N-terminally fused to SmBiT and LgBiT, respectively, with a 15-amino acid flexible linker (GGSGGGGSGGSSGG). Similarly, the human Cdc42 (residues 2–191) and the GBD of the human WAS (residues 220–288) were N-terminally fused to SmBiT and LgBiT, respectively, with the flexible linker. Coding sequence for Rac1, PAK1-GDB, Cdc42, and WAS-GBD were human codon-optimized and gene-synthesized by Genscript and inserted into the pCAGGS plasmid using a NEBuilder assembly kit. βarr2-Strawberry (*Thomsen et al., 2016*) was a gift from Prof. Larry Barak (Duke University, USA). Halo-miniGi (also referred to as miniGsi) was kindly provided by Prof. Nevin A. Lambert (Augusta University, USA). RFP-EEA1 (TagRFP-T-EEA1 cloned into pEGFP-C1 vector) and HA-Dyn-K44A (cloned into pcDNA3.1) constructs were gifts from Silvia Corvera and Sandra Schmid (respectively Addgene plasmids #42635 and #34683). RFP-Lck (C-tRFP-Lck cloned into PCMV6-AC-RFP expression vector) was purchased from Origene (#RC100049).

## Cell cultures and transfection

HEK293 cells (Life Technologies, cat. no. R705-07) were cultured in DMEM high glucose supplemented with 10% (v/v) fetal bovine serum (FBS) and 100 U/ml penicillin/streptomycin (1% (v/v) P/S). Jurkat T cells (ATCC, cat. no. TIB-152, clone E6-1) were cultures in RPMI 1640 media supplemented with 10% FBS and 1% P/S. The cell lines were authenticated by the supplier using STR profiling and tested negative for mycoplasma contamination.

DNA for BRET or confocal imaging experiments to be transfected was combined with salmon sperm DNA to obtain a total of 1 µg DNA/condition. PEI was combined with DNA and incubated 20 min before being added to the cells. DNA constructs for all other experiments were transfected into the cells using Lipofectamine 3000.

## Stable cell line construction

Transfected and construct-expressing HEK293 cells were placed in culture media containing 500 µg/mL G418 (HEK293-CCR7), 2 µg/mL puromycin (HEK293-CCR7-APEX2), 100 µg/ml zeocin (HEK293-CCR7 cells used to generate cell lines stably expressing PM-APEX2, ENDO-APEX2, or CYTO-APEX2), or 100 µg/mL zeocin +100 µg/mL G418 (HEK293-CCR7-PM-APEX2, HEK293-CCR7-ENDO-APEX2, or HEK293-CCR7-CYTO-APEX2) to initiate selection of stably transfected cells. Individual clones/colonies were expanded and functional expression of constructs was verified by fluorescence imaging (PM-APEX2, ENDO-APEX2, or CYTO-APEX2), or CCL19-mediated inhibition of forskolin-stimulated cAMP production (CCR7 and CCR7-APEX2).

## CAMYEL real-time cAMP assay

The CAMYEL biosensor (*Jiang et al., 2007*) (YFP-Epac-Rluc) transfected cells were equilibrated in HBSS supplemented with 10 mM HEPES (pH 7.4) at 37 °C for 30 min. Coelenterazine-h was added at a final concentration of 5 µM before starting the measurement. After establishing a baseline response for 2 min, cells were stimulated with forskolin at a final concentration of 10 µM and the response was measured. Five minutes after the forskolin stimulation, CCL19 or CCL21 was added at a final concentration of 100 nM and the luminescence was measured for further 15 min. The signal was detected at 550 nm using a CLARIOstar instrument (BMG LabTech). No blinding from the different conditions was done.

## BRET-based assays

Transfected cells were washed with DPBS and assayed in Tyrode's buffer (137 mM NaCl, 0.9 mM KCl, 1 mM MgCl$_2$, 11.9 mM NaHCO$_3$, 3.6 mM NaH$_2$PO$_4$, 25 mM HEPES, 5.5 mM glucose, 1 mM CaCl$_2$, pH 7.4) at 37 °C. 100 nM CCL19, or 100 nM CCL21, or vehicle were added to cells and incubated at 37 °C for 30 min. Five minutes before reading, Renilla luciferase II (RlucII) substrate (coelenterazine 400 a) was added at a final concentration of 2.5 µM. All BRET measurements were performed using a FLUOstar Omega microplate reader (BMG Labtech) with an acceptor filter (515±30 nm) and donor

filter (410±80 nm). BRET was calculated by dividing GFP emission by RlucII emission. No blinding from the different conditions was done.

## DiscoverX PathHunter β-arrestin assay

The PathHunter protein complementation assay (DiscoverX) using the PathHunter HEK 293 CCR7 β-Arrestin cell line was performed according to the manufacturer's protocol using and read for chemiluminescent signaling on a NovoStar plate reader (BMG Labtech). No blinding from the different conditions was done.

## NanoBiT assays

Multiple NanoBiT assays were performed between CCR7-SmBiT/LgBiT-CAAX, CCR7-SmBiT/LgBiT-FYVE, LgBiT-miniG/SmBiT-βarr1, LgBiT-RhoA/SmBiT-PKN1-GDB, LgBiT-PAK1-GBD/SmBiT-Rac1, and LgBiT-WAS1-GDB/SmBiT-Cdc42 to check the interaction between each biosensor pair. NanoBiT biosensor transfected cells were equilibrated in Opti-MEM at 37 °C for 60 min. Coelenterazine-h was added at a final concentration of 10 μM before starting the measurement. After establishing a baseline response for 2 min, cells were stimulated with CCL19 or CCL21 added at a final concentration of 100 nM and the luminescence was measured for further 20 min. The signal was detected at 550 nm using a PHERAstar *FSX* instrument (BMG LabTech). No blinding from the different conditions was done.

## Confocal microscopy

Confocal microscopy experiments were conducted on CCR7 expressing HEK293 cells co-transfected with Halo-miniGi/βarr2-Strawberry, Halo-miniGi/RFP-Lck, or Halo-miniGi/RFP-EEA1. On the same day as the experiments, Oregon green Halo Tag ligand was added to Halo-miniGi expressing cells at a final concentration of 1 μM in the media and incubated 15 min at 37 °C. Cells were washed three times with the media and incubated 30 min for the last wash at 37 °C. The media was then aspirated, replaced by Tyrode's buffer and 100 nM CCL19, 100 nM CCL21, or vehicle were added and the cells were incubated for 30 min at 37 °C. Cells were fixed with 4% paraformaldehyde in PBS for 10 min at room temperature, washed with DPBS, incubated 10 min in DPBS, and then DPBS was replaced by Tyrode's buffer. Cells were visualized using a Leica SP8 confocal microscope and analyzed using Leica Application Suite X (LASX). The same microscope was used to determine correct subcellular localization of the PM-APEX2, ENDO-APEX2, and CYTO-APEX2 proteins in HEK293 cells. No blinding from the different conditions was done.

## Co-localization analysis

Co-localization was performed using Imaris software version 9.9.1 (Bitplane, Oxford instruments, Switzerland). Cells were selected using the 'surfaces' module and cells containing regions of interest were selected. Channels for each fluorophore within the selected cells were masked to exclude all other cells or noise within the image. The Coloc module was then used to calculate co-localized voxels between the different channels. Thresholds were selected based on levels of intensity. Data are reported as a percentage of red volume (red voxels) above the threshold that is co-localized with green volume (green voxels) above the threshold. No blinding from the different conditions was done.

## Cisbio cAMP d2 dynamic assay

HEK293-CCR7-APEX2 or Jurkat T cells were harvested and resuspended in assay buffer (500 nM IBMX +20 mM HEPES in HBSS buffer, pH 7.4). In a white small volume 384-well plate, 5 μL of ligand buffer (assay buffer +20 μM forskolin) containing 200 nM CCL19, 200 nM CCL21, or vehicle was mixed with 5 μL of cell suspension. The plate was incubated at room temperature for 30 min. Next, 10 μL of lysis buffer containing 1.25% $Eu^{3+}$-anti-cAMP antibody and 1.25% cAMP-d2 was added and the plate was incubated for 1 hr at room temperature. The plate was read on a PHERAstar FSX instrument (BMG LabTech) where wells were excited with light at 340 nm and emission light was measured at 615 nm and 665 nm. The TR-FRET 665 nm/615 nm ratio, which is inversely proportional to the cAMP production, was used in combination with a cAMP standard curve to calculate the cAMP production in the cells. No blinding from the different conditions was done.

## Western blotting

To confirm correct biotinylation in HEK293-CCR7-APEX2, HEK293-CCR7-PM-APEX2, HEK293-CCR7-ENDO-APEX2, or HEK293-CCR7-CYTO-APEX2 cell lines, western blotting was conducted using streptavidin-Alexa488. All biotinylated proteins were visualized on an Amersham Typhoon instrument (Cytiva).

To confirm the expression of HA-tagged Dynamin-K44A mutant (HA-Dyn-K44A) or β-tubulin, western blotting was performed using a mouse anti-HA antibody followed by donkey anti-mouse IgG antibody conjugated to Alexa Fluor 488 or rabbit anti-β-tubulin antibody followed by donkey anti-rabbit IgG F(ab′)2 conjugated to HRP. Protein bands representing HA-Dyn-K44A were visualized on an Amersham Typhoon instrument (Cytiva) while bands corresponding to β-tubulin was visualized in ChemiDoc XRS+ (Bio-Rad) using SuperSignal West Pico PLUS Chemiluminescent Substrate. No blinding from the different conditions was done.

## ELISA cell surface expression

Cell surface expression of CCR7-APEX2 was evaluated using ELISA. HEK293 cells or HEK293 cells stably expressing CCR7-APEX2 were seeded in poly-D-lysine coated 96-well white plates. Cells were fixed with 4% paraformaldehyde and blocked with 2% BSA in PBS, then incubated 1 hr at room temperature with Anti-FLAG M2-HRP conjugated antibody diluted 1:10,000 in blocking buffer. Enzymatic chemiluminescence was generated by addition of SuperSignal West Pico PLUS as substrate and measured on a CLARIOstar plate reader (BMG LabTech) with the emission filter 410–80.

## Cell migration

Jurkat T cells suspended in culture medium were transferred into Costar Transwell polycarbonate membrane inserts with 5 µm pore size (Millipore Sigma). Each insert was fitted into a well in a 24-well plate containing culture media with either vehicle control solution, 100 nM CCL19, or 100 nM CCL21. Cells were allowed to migrate from the top insert to the lower well chamber of the plate for 2 hr at 37 °C at 5% $CO_2$. At the end of the experiment, all migrated cells from the lower chamber were collected and transferred to a Countess Cell Counting Chamber Slide (Thermo Fisher Scientific) and counted using a Countess 3 Automated Cell Counter (Thermo Fisher Scientific). Cells used for assessment of $G_{i/o}$ protein involvement in CCR7-stimulated migration, were pretreated with 100 ng/ml PTX or vehicle control for 16 hr prior to the experiment. Cells used for assessment of the dependency of receptor internalization in CCR7-stimulated chemotaxis, were pretreated with 30 µM Dyngo-4a or inactive Dyngo control compound for 30 min prior to the experiment. In additional experiments, the cells were pre-incubated with 10 µM PitStop2 or inactive PitNot control compound for 30 min prior to the experiment. Cells used for assessment of Rac1's role in CCR7-stimulated chemotaxis were pretreated with 10 µM EHT1864 or DMSO control for 30 min prior to the experiment. All chemotaxic responses were normalized to passive cell migration towards the control as 0% and CCL19 as 100% response. No blinding from the different conditions was done.

## Data and statistical analysis

All graphs were generated and analyzed using GraphPad Prism 8 (GraphPad Software). Data are presented as mean ± SEM and N refers to the number of independent experiments (or biological replicates) that have been conducted. Differences were assessed using Student's $t$-test for two comparisons and one- or two-way ANOVA and Tukey's post hoc test for multiple comparisons. $p<0.05$ was considered significant at the 95% confidence level.

## APEX2 labeling

CCR7-APEX2 transfected HEK293 cells were preincubated with 500 µM biotin-tyramide for 1 hr at 37 °C. 100 nM CCL19, 100 nM CCL21, or vehicle were added to stimulate cells for 0, 2, 10, or 25 min. The proximity labeling was initiated by addition of freshly diluted $H_2O_2$ from a 30% (v/v) stock solution to a final concentration of 1 mM. The culture media was removed exactly 1 min later, and the cells were washed three times with ice cold quenching buffer (PBS supplemented with 10 mM sodium ascorbate, 10 mM sodium azide, and 5 mM Trolox). The cells were detached by adding 5 mL of the quenching buffer supplemented with 5 mM EDTA and incubating at room temperature for 10 min with

agitation. Detached cells were transferred to a 15 mL conical tube and centrifuged at 1000×g at 4 °C for 10 min to remove the supernatant. The pellet was stored at –80 °C until cell lysis.

The frozen cell pellet was lysed with 1 mL RIPA buffer (50 mM Tris-HCl, 150 mM NaCl, 1% Triton X-100, 0.5% sodium deoxycholate, 0.1% SDS, pH 7.4) supplemented with 1 mM PMSF for 30 min at room temperature with agitation. The cell lysate was further solubilized by brief sonication and centrifuged at 12,000×g at 4 °C for 10 min to remove any insoluble cell debris. NeutrAvidin agarose beads were pre-washed three times with 20 bed volumes of the RIPA buffer and incubated with the supernatant overnight at 4 °C with agitation. After the incubation, the beads were washed by incubating at room temperature for 10 mins each with 40 bed volumes of buffer A (2% SDS), buffer B (50 mM HEPES, 500 mM NaCl, 1 mM EDTA, 1% Triton X-100, 0.1% sodium deoxycholate, pH 7.4), then buffer C (10 mM Tris-HCl, 500 mM NaCl, 1 mM EDTA, 0.1% Tween-20, 0.5% sodium deoxycholate, pH 8.0) with agitation. The beads were further washed twice with 20 bed volumes of 100 mM HEPES (pH 7.4) and once with PBS (pH 7.4) to remove trace amounts of detergent. Washed beads were resuspended in 100 µL PBS and subjected to mass spectrometry analysis. No blinding of the different conditions was done.

## Mass spectrometry and data acquisition

Proteins on the NeutrAvidin agarose beads were eluted by incubating with 50 µL of 50 mM Tris-HCl, 5% SDS, 10 mM TCEP, 20 mM CAA (pH 7.4) at 90 °C for 20 min. Eluted proteins were purified and digested on magnetic beads following SP3 workflow (*Hughes et al., 2019*). Eluates were transferred into clean Eppendorf tubes and mixed with SP3 magnetic beads. Proteins were precipitated by mixing with an equal volume of ethanol and SP3 beads were washed three times with 200 µL of 85% ethanol to remove any traces of SDS from the elution buffer. The SP3 beads were suspended in 50 µL of 50 mM Tris-HCl (pH 7.4) and the proteins on the beads were digested with trypsin (4 ng/µL) at 37 °C overnight. Trypsin digestion was quenched by adding TFA to a final concentration of 0.5%. Peptide digests were transferred from the magnetic beads into the clean tubes. SP3 beads were washed with 50 uL of 5% ACN 0.2% TFA. The peptides were loaded on Evotips C18 (Evosep) for subsequent LC-MS/MS analysis using Evosep One LC system coupled to Q Exactive HF-X Hybrid Quadrupole-Orbitrap MS instrument (Thermo Scientific) operating in data-independent acquisition mode (DIA). Peptides were separated online utilizing 15SPD method (88 min LC gradient length). High-resolution full MS1 spectra were acquired with a resolution of 120,000, automatic gain control (AGC) target at 3,000,000, and maximum ion injection time at 60 ms, with a scan range of 350–1650 m/z. Following each full MS1 scan, 22 data-independent high energy collisional dissociation (HCD) MS/MS scans were acquired at the resolution of 30,000, AGC target of 3,000,000 with stepped normalized collision energy (NCE) of 22.5, 25, and 27.5.

Collected DIA data were analyzed using Spectronaut software (Biognosis; https://biognosys.com/shop/spectronaut) and searched in directDIA mode against the SwissProt subset of the human UniProt database (http://www.uniprot.org/). Database search was performed in integrated search engine Pulsar (Biognosis). For the database search, the enzyme specificity was set to trypsin with the maximum number of missed cleavages allowed set to two. Oxidation of methionine was searched as variable modification, whereas carbamidomethylation of cysteines was searched as a fixed modification. The false discovery rate (FDR) for peptide, protein, and site identification was set to 1%. Protein quantification was performed on MS2 level using three most intense fragment ions per precursor. Proteins identified with a single peptide was removed from further analysis. The mass spectrometry raw files are accessible under MassIVE ID: MSV000090362 at https://massive.ucsd.edu. Subsequent data analysis steps were performed in Perseus and GraphPad Prism.

## Statistical analysis of the MS data

For spatiotemporal characterization of CCR7 upon agonist stimulation, global median normalization was performed for dataset from each time points (0, 2, 10, and 25 min after agonist stimulation). Pairwise differential expression analysis was performed by comparing the stimulated datasets (2, 10, and 25 min) to the unstimulated (0 min) dataset using t-test model, assuming equal variance for each protein being compared. For each stimulated time point, proteins with p-value <0.05 and $\log_2$(fold-change) >1 were considered significant. Significance score for the proteins were calculated by taking the absolute value of the $\log_2$(fold-change) and adding $-\log_{10}$(p-value) (*Xiao et al., 2014*).

Proteins with relevance to Rho, Rac, or CDC42 were selected manually from the list and subjected to further analysis for the role of CCR7 in cell motility. These proteins were searched on the STRING database to build a network functional relevance.

For spatial references PM-APEX2, ENDO-APEX2, and CYTO-APEX2, global median normalization was also performed. Pairwise differential expression analysis between each pair (ENDO against PM, ENDO against CYTO, and PM against CYTO) were performed as above. For each comparison pair, proteins with p-value <0.05 and $\log_2$(fold-change) >1 were considered significant. Then, proteins commonly appear significant for ENDO and PM from the $t$-tests were chosen as spatial references.

## Acknowledgements

We are grateful to Dr. Nigel Bunnett for fruitful discussions and generous gift of the plasmid encoding the CAMYEL cAMP biosensor. We thank Dr. Mark von Zastrow for the generous gifts of the plasmids encoding PM-APEX2, ENDO-APEX2, and CYTO-APEX2. We are also grateful for support by and discussions with Evgeny Kanshin and Beatrix Ueberheide at NYU School of Medicine Proteomics Laboratory. Funding:This work received support from the LEO Foundation (LF18043 to ARBT), NIH grants (R35GM147088 (NIGMS) and R21CA243052 (NCI) to ARBT), a Wellcome Trust Seed Award (215229/Z/19/Z to BP), and a BBSRC New Investigator Award (BB/X002578/1 to BP). AI was funded by Japan Society for the Promotion of Science (JSPS) KAKENHI grants JP21H04791, JP21H05113, JP24K21281, JPJSBP120213501 and JPJSBP120218801; FOREST Program JPMJFR215T and JST Moonshot Research, and Development Program JPMJMS2023 from Japan Science and Technology Agency (JST); The Uehara Memorial Foundation; and Daiichi Sankyo Foundation of Life Science.

## Additional information

### Competing interests

Alex RB Thomsen: a founding scientist of Unco Therapeutics LLC. The other authors declare that no competing interests exist.

### Funding

| Funder | Grant reference number | Author |
|---|---|---|
| LEO Fondet | LF18043 | Alex RB Thomsen |
| National Institute of General Medical Sciences | R35GM147088 | Alex RB Thomsen |
| National Cancer Institute | R21CA243052 | Alex RB Thomsen |
| Wellcome Trust | 10.35802/215229 | Bianca Plouffe |
| Biotechnology and Biological Sciences Research Council | BB/X002578/1 | Bianca Plouffe |
| Japan Society for the Promotion of Science | JP21H04791 | Asuka Inoue |
| Japan Society for the Promotion of Science | JP21H05113 | Asuka Inoue |
| Japan Society for the Promotion of Science | JP24K21281 | Asuka Inoue |
| Japan Society for the Promotion of Science | JPJSBP120213501 | Asuka Inoue |
| Japan Society for the Promotion of Science | JPJSBP120218801 | Asuka Inoue |
| Japan Science and Technology Agency | 10.52926/JPMJFR215T | Asuka Inoue |

| Funder | Grant reference number | Author |
|---|---|---|
| Japan Science and Technology Agency | 10.52926/JPMJMS2023 | Asuka Inoue |
| Uehara Memorial Foundation | | Asuka Inoue |
| Daiichi Sankyo Foundation of Life Science | | Asuka Inoue |

The funders had no role in study design, data collection and interpretation, or the decision to submit the work for publication. For the purpose of Open Access, the authors have applied a CC BY public copyright license to any Author Accepted Manuscript version arising from this submission.

## Author contributions

Hyunggu Hahn, Carole Daly, John Little IV, Conceptualization, Data curation, Formal analysis, Investigation, Methodology, Writing – review and editing; Nicole A Perry-Hauser, Formal analysis, Investigation, Writing – review and editing; Emmanuel Flores-Espinoza, Formal analysis, Investigation, Methodology; Asuka Inoue, Resources, Funding acquisition, Methodology, Writing – review and editing; Bianca Plouffe, Alex RB Thomsen, Conceptualization, Resources, Data curation, Formal analysis, Supervision, Funding acquisition, Validation, Investigation, Visualization, Methodology, Writing – original draft, Project administration, Writing – review and editing

## Author ORCIDs

Hyunggu Hahn ⬤ https://orcid.org/0000-0003-3532-7671
Nicole A Perry-Hauser ⬤ https://orcid.org/0000-0003-3130-3023
Emmanuel Flores-Espinoza ⬤ https://orcid.org/0000-0002-7889-7458
Asuka Inoue ⬤ https://orcid.org/0000-0003-0805-4049
Bianca Plouffe ⬤ https://orcid.org/0000-0002-8321-0796
Alex RB Thomsen ⬤ https://orcid.org/0000-0003-1638-8911

Reviewer #1 (Public review): https://doi.org/10.7554/eLife.99373.3.sa1
Reviewer #2 (Public review): https://doi.org/10.7554/eLife.99373.3.sa2
Author response https://doi.org/10.7554/eLife.99373.3.sa3

# Additional files

## Supplementary files
MDAR checklist

## Data availability
The mass spectrometry raw files are accessible under MassIVE ID: MSV000090362 at https://massive.ucsd.edu.

The following dataset was generated:

| Author(s) | Year | Dataset title | Dataset URL | Database and Identifier |
|---|---|---|---|---|
| Hahn et al. | 2025 | Endosomal Chemokine Receptor Signalosomes Regulate Central Mechanisms Underlying Cell Migration | https://massive.ucsd.edu/ProteoSAFe/dataset.jsp?accession=MSV000090362 | MaasIVE, MSV000090362 |

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
