## [Editor Report · eLife Assessment]

This is an **important** study that provides CCR7-APEX2 proximity labelling mass spectrometry data that is expected to provide new insights into CCR7 signaling partners and pathways. The study is technically easy to follow and the data is **convincing**. It will be interesting in the future to have complementary studies in lymphocytes/dendritic cells that endogenously express CCR7. This is of value to the community, and there are likely multiple opportunities to use the APEX2 data set to extend these findings, strengthen some claims, and even explore a new pathway identified in the APEX2 data set.

---

## [Referee Report · Reviewer #1 (Public review)]

Summary:

Hahn et al use bystander BRET, NanoBiT assays and APEX2 proteomics to investigate endosomal signaling of CCR7 by two agonists, CCL19 and CCL21. The authors suggest that CCR7 signals from early endosomes following internalisation. They use spatial proteomics to try to identify novel interacting partners that may facilitate this signaling and use this data to specifically enhance a Rac1 signaling pathway. The most novel findings are the APEX2 proteomics studies that provide new mechanisms.

Strengths:

(1) The APEX2 resource will be valuable to the GPCR and immunology community. It offers many opportunities to follow up on findings and discover new biology. The authors have used the resource to validate earlier findings in the current manuscript and in previous manuscripts.

(2) The results section is well written and can be followed very easily by the reader.

(3) Some findings verify previous studies (e.g. endomembrane signalling).

Weaknesses:

(1) The findings are interesting although the studies are almost all performed in HEK293 cells. I understand that these are commonly used in GPCR biology and current tools need to be improved in order to perform similar analyses in more relevant cell-lines. Future studies should focus on validating the findings of the current study in physiologically-relevant cell-lines.

---

## [Referee Report · Reviewer #2 (Public review)]

Summary:

This manuscript describes a comprehensive analysis of signalling downstream of the chemokine receptor CCR7. A comprehensive dataset supports the authors' hypothesis that G protein and beta arrestin signalling can occur simultaneously at CCR7 with implications for continued signalling following receptor endocytosis.

Strengths:

The experiments are well controlled and executed, employing a wide range of assay, using in the main, CCR7 transfectants. Data are well presented, with the authors claims supported by the data. The paper also has an excellent narrative which makes it relatively easy to follow. I think this would certainly be of interest to the readership of the journal.

Weaknesses:

The experiments are currently representative of signalling events in HEK293 transfectants and await verification in more relevant systems e.g. T-cells and dendritic cells.

Appraisal and Discussion

Overall, the authors appear to have achieved their experimental aims and provide substantial evidence that chemokine receptors can stimulate G proteins from within endosomes to regulate signalling pathways involved in cell migration. This builds upon earlier studies from the Legler group which showed that endocytosed CCR7 could activate Rac1 and influence lamellipodia formation. An unbiased mass spectrometry-based proteome profiling approach was used by the authors of this study to identify several candidate proteins which appear to play a role in receptor trafficking and signalling downstream of CCR7. These data may provide clues as to how other chemokine receptors are regulated post endocytosis in various leukocyte subsets.

---

## [Author Response]

The following is the authors’ response to the original reviews.

**Public Reviews:**

**Reviewer #1 (Public Review):**
Summary:Hahn et al use bystander BRET, NanoBiT assays, and APEX2 proteomics to investigate endosomal signaling of CCR7 by two agonists, CCL19 and CCL21. The authors suggest that CCR7 signals from early endosomes following internalisation. They use spatial proteomics to try to identify novel interacting partners that may facilitate this signaling and use this data to specifically enhance a Rac1 signaling pathway. Many of the results in the first few figures showing simultaneous recruitment of Barr and G proteins by CCR7 have been shown previously (Laufer et al, 2019, Cell Reports), as has signaling from endomembranes, and Rac1 activation at intracellular sites. The new findings are the APEX2 proteomics studies, which could be useful to the scientific community. Unfortunately, the authors only follow up on a single finding, and the expansion of this section would improve the manuscript.

First of all, we would like to thank the reviewer for helping with the manuscript. The summary is mostly accurate except for the statement that simultaneous recruitment of barr and G protein to CCR7 has been shown before. It should also be noted that it has not been demonstrated that CCR7 activates G proteins from endosomes previously nor has the functional role of this signaling mechanism. However, that CCR7 activity at endomembranes is associated with Rac1 signaling was demonstrated in the Laufer et al. study as the reviewer correctly points out.

Strengths:(1) The APEX2 resource will be valuable to the GPCR and immunology community. It offers many opportunities to follow up on findings and discover new biology. The resource could also be used to validate earlier findings in the current manuscript and in previous manuscripts. Was there enrichment of early endosomal markers, Barr and Gi as this would provide further evidence for their earlier claims regarding endosomal signaling? Previous studies have suggested signaling from the TGN, so it is possible that the different ligands also direct to different sites. This could easily be investigated using the APEX2 data.

Thank you for your comment. We do in fact observe enrichment of TGN/Golgi markers in response to chemokine stimulation, which we now have highlighted in the manuscript (fourth paragraph on page 7).

(2) The results section is well written and can be followed very easily by the reader.

We are glad that the reviewer found the results section very readable.

(3) Some findings verify previous studies (e.g. endomembrane signalling). This should be acknowledged as this shows the validity of the findings of both studies.

This is correct. We have now included more discussion of previous work related to CCR7 signaling at endomembranes (thirdparagraph on page 10).

Weaknesses:(1) The findings are interesting although the studies are almost all performed in HEK293 cells. I understand that these are commonly used in GPCR biology and are easy to transfect and don't express many GPCRs at high concentrations, but their use is still odd when there are many cell-lines available that express CCR7 and are more reflective of the endogenous state (e.g. they are polarised, they can perform chemotaxis/ migration). Some of the findings within the study should also be verified in more physiologically relevant cells. At the moment only the final figure looks at this, but findings need to be verified elsewhere.

We thank the reviewer for raising this point and giving us an opportunity to elaborate in further detail. The major goal of our study was to investigate whether CCR7 activates G protein from endosomes, the underlying mechanism, and functions of this potential signaling mechanism. The reason we chose CCR7 as our model receptor was that it belongs to a group of GPCRs, the chemokine receptors, that most often have features associated with the ability to promote endosomal G protein activation (phosphorylation site clusters in the C-terminal region).

Specific detection of G protein activation at distinct subcellular compartments is currently very challenging in truly endogenous systems despite new innovative biosensors that are available (not just related to CCR7, but GPCRs in general). To our knowledge, most if not all studies that detect direct activation of G protein at a specific compartment whether at the plasma membrane, endosome, Golgi, or other compartments, have overexpressed either the receptor, G protein, or both. This is why we choose the HEK293 cell system for most of our experiments, which are easy to manipulate. That being said, we did confirm major findings in an indirect manner using Jurkat T-cells, which express CCR7 endogenously and are physiological relevant. Our hope is that in the future we will be able to use highly sensitive biosensors to directly confirm our findings in such a cell system as the reviewer wisely suggests.

(2) The authors acknowledge that the kinetic patterns of the signals at the early endosome are not consistent with the rates of internalisation. They mention that this could be due to trafficking elsewhere. This could be easily looked at in their APEX2 data. Is there evidence of proximity to markers of other membranes? Perhaps this could be added to the discussion. Similarly, previous studies have shown that CCR7 signaling may involve the TGN. Was there enrichment of these markers? If not, this could also be an interesting finding and should be discussed. It is also possible that the Rab5 reporter is just not as efficient as the trafficking one, especially as in later figures the very convincing differences in the two ligands are not as robust as the differences in trafficking.

Excellent point. We have now highlighted the possibility of CCR7 being further trafficked to the trans-Golgi network (TGN) as possible explanation for the transient translocation of activated CCR7 to the early endosome in Fig. 1G-H (second paragraph on page 3).

Furthermore, in the APEX2 experiment we observe enrichment of proteins involved in lysosomal trafficking (LAMP1, VPS16, VAMP7, WDR91, and PP4P1) by CCL19 stimulation at 25 min, and recycling endosomes/TGN markers (SNX6, RAB7L, and GGA) by CCL21 stimulation at 25 min. In addition to this, several markers of TGN/Golgi (SNX3, COG5, YIF1A, SC22B, and AP3S1) were enriched as well in response to both CCL19 and CCL21 stimulation. We have now included a statement in the manuscript, which describes the likely trafficking of CCR7 to the TGN/Golgi in response to CCL19 and CCL21 stimulation (fourth paragraph on page 7).

(3) In the final sentence of paragraph 2 of the results the authors state that the internalisation is specific to CCR7 as there isn't recruitment to V2R. I'm not sure this is the best control. The authors can only really say it doesn't recruit to unrelated receptors. The authors could have used a different chemokine receptor which does not respond to these ligands to show this.

The point with this control experiment was to demonstrate that the loss of NanoBiT signal in response to CCL19 in CCR7-SmBiT/LgBiT-CAAX expressing cells, but not in V2R-SmBiT/LgBiT-CAAX expressing cells, was a result of bona fide CCR7 internalization rather than potential artifactual effects of CCL19 on the NanoBiT system. Our intent was not to demonstrate specificity of CCL19 among chemokine receptors, which already has been thoroughly tested in previous studies. We have now modified the sentence (second paragraph on page 3) “Moreover, CCL19/CCL21-stimulation of receptor internalization to endosomes is specific to CCR7 as none of the chemokines promote internalization or trafficking to endosomes of the vasopressin type 2 receptor (V_2_R)-SmBiT construct (Fig. S1E-F)” to “Moreover, CCL19/CCL21-stimulation did not promote internalization or trafficking to endosomes of the vasopressin type 2 receptor (V_2_R)-SmBiT construct, which validates that these chemokines act specifically via the CCR7-SmBiT system (Fig. S1E-F).”

(4) The miniGi-Barr1 and imaging showing co-localisation could be more convincing if it was also repeated in a more physiological cell line as in the final figure. Imaging of CCR7, miniGi, and Barr1 would also provide further evidence that the receptor is also present within the complex.

We agree with the reviewer’s assessment. However, as mentioned above it is currently extremely challenging to detect endogenous G protein coupling/activation to endogenous receptors. In addition, we are not sure if overexpressing fluorophore-tagged receptor, miniG, and barr1 in a physiological-relevant cell line would provide truly physiological conditions as the expression of these proteins still would be artificially high. This is why we chose to conduct these mechanistic experiments in HEK293 cells and then indirectly verify key findings in an endogenous and physiological-relevant cell line.

(5) The findings regarding Rac1 are interesting, although an earlier paper found similar results (Laufer et al, 2019, Cell Reports), so perhaps following up on another APEX2-identified protein pathway would have been more interesting. The authors' statement that Rac1 is specifically activated, and RhoA and Cdc42 are not, is unconvincing from the current data. Only a single NanoBiT assay was used, and as raw values are not reported it is difficult for the reader to glean some essential information. The authors should show evidence that these reporters work well for other receptors (or cite previous studies) and also need evidence from an independent (i.e. non-NanoBiT or BRET) assay.

The major focus of the study was to investigate whether CCR7 can activate G protein after having been internalized into endosomes via formation of CCR7-Gi/o-barr megaplexes, and to dissect out potential functions of said endosomal G protein signaling. To do this, we used CCL19 and CCL21 which stimulate G protein to the same extent but differ in their ability of promote barr recruitment and receptor internalization with CCL19 being superior to CCL21. To this end, we found that CCL19 also promote endosomal G protein activation to a greater extent than CCL21, and therefore, we specifically looked for proteins enriched by CCL19 in our APEX experiment. This led us to some Rho GTPase regulators that were differentially enriched by CCL19 and CCL21. We agree that there were other interesting effectors related to CCR7 biology identified in the APEX experiment such as EYA2, GRIP2, and EI24. However, those proteins were enriched similar by CCL19 and CCL21 challenge, and thus, do not seem to be activated specifically at endosomes. Following the same argument, we also did not observe any difference in the activity of RhoA or Cdc42 when stimulated with CCL19 or CCL21, so we cannot conclude that these signaling proteins are activated specifically in endosomes. On the other hand, Rac1 was stimulated to a larger degree by CCL19 than CCL21, its activity was inhibited by the Gi/o inhibitor PTX and endocytosis inhibitors Dyngo-4a and PitStop2. CCR7-mediated Rac1 signaling was also inhibited by expression of a dominant negative dynamin mutant that inhibits receptor internalization, and Rac1 was not activated by an internalization-deficient CCR7-DS/T mutant. Finally, the involvement of Rac1 in CCR7 mediated chemotaxis of Jurkat T cells was also demonstrated. We believe that these findings together provide strong basis for the claim that endosomal Gi/o protein signaling by CCR7 activates Rac1.

Following the reviewer’s suggestion, we have now included experiments to show that the activation of RhoA, Rac1, and Cdc42 by CXCR4 also can be detected by the NanoBiT biosensors (Fig. S7D-F). We have also added the appropriate references to the original studies where these biosensors were developed in the results section (first paragraph on page 8).

(6) At present, the studies in Figure 7 do not go beyond those in the previous Laufer et al study in which they showed blocking endocytosis affected Rac1 signalling. The authors could show that Rac1 signalling is from early endosomes to improve this, otherwise, it could be from the TGN as previously reported.

The major purpose of Figure 7 was to indirectly confirm findings from HEK293 cells experiments and to tie them to physiological functions. Our experiments using Jurkat T-cells show that CCL19 promote stronger chemotactic response than CCL21 despite similar Gi/o response. In addition, we showed that CCR7-mediated Gi/o activation, receptor endocytosis, as well as Rac1 activity, are required to drive chemotaxis. The Laufer et al. study did not investigate whether CCR7 activates G protein after having been internalized into endosomes via formation of CCR7-Gi/o-barr megaplexes, and thus, did not focus on functional outcomes of this signaling mechanism. Based on this, we believe our work provides new and valuable knowledge to the field.

**Reviewer #2 (Public Review):**
Summary:This manuscript describes a comprehensive analysis of signalling downstream of the chemokine receptor CCR7. A comprehensive dataset supports the authors' hypothesis that G protein and beta-arrestin signalling can occur simultaneously at CCR7 with implications for continued signalling following receptor endocytosis.

We would like to thank the reviewer for helping with the manuscript. We agree on all points made and have now updated the manuscript accordingly.

Strengths:The experiments are well controlled and executed, employing a wide range of assays using - in the main - CCR7 transfectants. Data are well presented, with the authors' claims supported by the data. The paper also has an excellent narrative which makes it relatively easy to follow. I think this would certainly be of interest to the readership of the journal.

We appreciate the positive assessment of strengths.

Weaknesses:Since the authors show a differential enrichment of RhoGTPases by CCR7 stimulation with CCL19 versus CCL21, I think that they also need to show that the Gi/o coupling of HEK-292-CCR7-APEX2 cells to both CCL19 and CCL21 is not perturbed by the modification. Currently, the authors only show data for CCL19 signalling, which leaves the potential for a false negative finding in terms of CCL21 signalling being selectively impaired. This should be relatively easy to do and should strengthen the authors' conclusions.

We agree with the reviewer and have now included experiments to show that both CCL19- and CCL21-mediated CCR7-APEX2 stimulation leads to Gi/o activation (Fig. S4C). In addition, our proteomics experiments show strong effects of both CCL19 and CCL21 stimulation, which suggest that the receptor is activated by both ligands.

The authors conclude the discussion by suggesting that their findings highlight endosomal signalling as a general mechanism for chemokine receptors in cell migration. I think this is an overreach. The authors chose several studies of CXC chemokine receptors to support their argument that C-terminal truncation or mutation of the C-terminal phosphorylation sites impairs endocytosis and chemotaxis (refs 40-42). However, in some instances e.g. at the related chemokine receptor CCR4, C-terminal removal of these sites impairs endocytosis but promotes chemotaxis (Nakagawa et al, 2014; Anderson et al, 2020). I therefore think that either the final statement needs to be tempered down or the counterargument discussed a little.

We appreciate the reviewer highlighting this point. We have now modified the concluding sentence from “Thus, the findings from our study highlight endosomal G protein signaling by chemokine receptors as a potential general mechanism that regulates key aspects of cell migration” to “Thus, the findings from our study highlight endosomal G protein signaling by some chemokine receptors as a potential mechanism that regulates key aspects of cell migration.” We hope that the temper level of this sentence is more appropriate.

References:

Anderson, C. A. et al. A degradatory fate for CCR4 suggests a primary role in Th2 inflammation. J Leukocyte Biol 107, 455-466 (2020).

Nakagawa, M. et al. Gain-of-function CCR4 mutations in adult T cell leukaemia/lymphoma. Journal of Experimental Medicine 211, 2497-2505 (2014).

**Recommendations for the authors:**

**Reviewer #1 (Recommendations For The Authors):**
(1) The results section is well written, although the introduction needs more information on what is known about CCR7 trafficking and endomembrane signaling. I understand this is because the authors wanted to focus on GPCR signaling, but the study will equally be of interest to researchers in the immunology and chemokine fields, and therefore more CCR7-focussed discussion in the introduction would be useful. Similarly, the discussion would benefit from more discussion of previous studies of CCR7 trafficking and endomembrane signaling (in particular the Laufer et al paper) to acknowledge that many of the findings within this paper verify previous studies.

We have now included additional immunology/endomembrane background information about CCR7 at the place where the receptor is introduced (first paragraph on page 3). We have also expanded our discussion of our work in relation to the Laufer et al. study (third paragraph on page 10).

(2) On page 5, the authors state that 'The response to chemokine stimulation was not observed in mock transfected HEK293 cells'. Figure S4D does not have a legend so it is difficult to see what they mean by mock transfected. Do they mean not transfecting with anything or not with the receptor? The better control would be transfecting the reporters but not the receptor. This may have been done, but the wording needs clarifying and S4D needs a legend.

Thanks for pointing this out. We believe the reviewer refers to Figure S2D and we have now highlighted/clarified the legend better. Mock transfected conditions refer to HEK293 cells transfected with the reporter, but not the receptor. This is written in the legend as “(D) Change in luminescence signal generated between SmBiT-barr1 and LgBiT-miniGi in response to 100 nM CCL19 or 100 nM CCL21 in mock transfected HEK293 cells (no CCR7)”, which we believe should be clear to the audience.

(3) The validation of the APEX2 receptor construct relies on a single assay with one ligand. The authors should show that the receptor expresses at the cell surface, is internalised normally, and that both ligands activate the receptor.

We have now included additional data to show that (1) the receptor is expressed at the cell surface, (2) that the CCR7-APEX2 recruits barr1 to the plasma membrane, (3) that this association leads to barr1 translocation to the early endosomes as an indirect measurement of receptor internalization, and (4) that both CCL19- and CCL21-stimulation inhibit forskolin induced cAMP production (Fig.S4A-C, and described in fifth paragraph on page 6).

(4) The APEX2 section is very short, especially as this is novel data. It lacks some important information, e.g. when the authors state that 'we identified a total of 579 proteins', is this in total for both ligands, separately or were some shared? More information on each ligand separately and combined would make this clearer.

We have now specified that the identified total proteins enriched from our APEX2 approach is when the cells are stimulated with either CCL19 or CCL21 (third paragraph on page 7). Furthermore, we have included a Venn diagram in Fig. S5C to show how many proteins were enriched by CCL19 or CCL21 stimulation and how many of those were shared at different time points.

(5) The discussion would benefit from some further work. The current first two paragraphs just reiterate the introduction and don't discuss the current paper so could be removed completely. The Laufer et al study needs much more discussion as they report many of the findings of the current paper (signaling following endocytosis, Rac1 endomembrane signaling) five years ago. The APEX2 findings that are discussed, though interesting, are not followed up by further experimental evidence and there is little discussion of why the two ligands have different responses or what the physiological effects could be.

We appreciate the reviewer’s effort in helping with the discussion. To this end, we have now expanded our discussion of the mentioned paper further as suggested (third paragraph on page 10). We agree that the findings from our APEX experiment are interesting, but the focus of this study relates to proteins enriched specifically at endosomes. Several of the most enriched proteins did not show this localization bias, which is why these proteins were not further investigated.

Minor changes:(1) The authors should remove the word 'recent' at the start of the first sentence of the third paragraph. Endosomal signaling by GPCRs was described 15 years ago so cannot really be seen as recent anymore.

We have now adjusted the manuscript accordingly.

(2) Tukey defaulted to Turkey in some places.

We thank the reviewer for pointing out these typos, which now have been corrected.

**Reviewer #2 (Recommendations For The Authors):**
Minor Points:(1) ACKRs do not couple to G proteins so it is peculiar to see them in this table. I would limit the table to the conventional CCR1-10, CXCR1-6 and XCR1. The ligand for XCR1 is XCL1 which is absent from the table.

We have now modified the table accordingly.

(2) CCL19 (formerly known as ELC) has been long known to be a more efficacious and potent ligand in chemotaxis assays (Bardi et al, 2001). This earlier reference should be added to the citations in the preceding statement on page 10.

This is an important study showing that CCL19 is more efficacious than CCL21 in promoting chemotaxis and that this has been known for decades. We have now included the reference accordingly (reference 59 in second paragraph on page 11).

(3) Figure 6, Panel Q. I think the legends for CCR7 and CCR7 delta ST might be flipped.

We thank the reviewer for pointing out this error. We have now corrected the figure panel.

(4) Figure S5 (or 5) might benefit from simple Venn diagrams showing the numbers of differentially enriched proteins following treatment with the two ligands at different time points.

We have included a Venn diagram in Fig. S5C to show how many proteins were enriched by CCL19 or CCL21 stimulation and how many of those where shared.

Reference:

Bardi, G., Lipp, M., Baggiolini, M. & Loetscher, P. The T cell chemokine receptor CCR7 is internalized on stimulation with ELC, but not with SLC. European Journal of Immunology 31, 3291-3297 (2001).